# Why are Adaptive Methods Good for Attention Models?

**Jingzhao Zhang**
MIT
jzhzhang@mit.edu

**Sai Praneeth Karimireddy**
EPFL
sai.karimireddy@epfl.ch

**Andreas Veit**
Google Research
aveit@google.com

**Seungyeon Kim**
Google Research
seungyeonk@google.com

**Sashank Reddi**
Google Research
sashank@google.com

**Sanjiv Kumar**
Google Research
sanjivk@google.com

**Suvrit Sra**
MIT
suvrit@mit.edu

## Abstract

While stochastic gradient descent (SGD) is still the *de facto* algorithm in deep learning, adaptive methods like Clipped SGD/Adam have been observed to outperform SGD across important tasks, such as attention models. The settings under which SGD performs poorly in comparison to adaptive methods are not well understood yet. In this paper, we provide empirical and theoretical evidence that a heavy-tailed distribution of the noise in stochastic gradients is one cause of SGD's poor performance. We provide the first tight upper and lower convergence bounds for adaptive gradient methods under heavy-tailed noise. Further, we demonstrate how gradient clipping plays a key role in addressing heavy-tailed gradient noise. Subsequently, we show how clipping can be applied in practice by developing an *adaptive* coordinate-wise clipping algorithm (ACClip) and demonstrate its superior performance on BERT pretraining and finetuning tasks.

## 1 Introduction

Stochastic gradient descent (SGD) is the canonical algorithm for training neural networks [24]. SGD iteratively updates model parameters in the negative gradient direction and seamlessly scales to large-scale settings. Though a well-tuned SGD outperforms adaptive methods [31] in many tasks including ImageNet classification (see Figure 1a), certain tasks necessitate the use of *adaptive* variants of SGD (e.g., Adagrad [10], Adam [14], AMSGrad [23]), which employ adaptive learning rates. For instance, consider training an attention model [29] using BERT [9]. Figure 1e shows that in spite of extensive hyperparameter tuning, SGD converges much slower than Adam during BERT training.

In this work, we provide one explanation for why adaptivity can facilitate convergence with theoretical and empirical evidence. The significant hint that initializes our work comes from the distribution of the stochastic gradients. For Imagenet, the norms of the mini-batch gradients are typically quite small and well concentrated around their mean. On the other hand, the mini-batch gradient norms for BERT take a wide range of values and are sometimes much larger than their mean value. More formally, while the distribution of the stochastic gradients in Imagenet is well approximated by a Gaussian, the distribution for BERT seems to be *heavy-tailed*. Such observation leads us to the question: does adaptivity stabilize optimization under heavy-tailed noise?

We provide a positive answer to the above question by performing both theoretical and empirical studies of the convergence of optimization methods under heavy-tailed noise. In this setting, some of the stochastic gradients are much larger than the mean and can excessively influence the updates of SGD. This makes SGD unstable and leads to its poor performance. A natural strategy to stabilize the updates is to *clip* the magnitude of the stochastic gradients. We prove that indeed it is sufficient to ensure convergence even under heavy-tailed noise. Based on the analysis, we then motivate the design of a novel algorithm (ACClip) that outperforms ADAM on BERT related tasks. Specifically, we make the following contributions:

- We empirically show that in tasks on which Adam outperforms SGD (BERT pretraining), the noise in stochastic gradients is heavy-tailed. On the other hand, on tasks where traditionally SGD outperforms Adam (ImageNet training), we show that the noise is well concentrated.
- In section 3, we study the convergence of gradient methods under heavy-tailed noise condition where SGD's performance degrades and its convergence might fail. We then establish (with upper and lower bounds) the convergence of *clipped* gradient methods under the same condition and prove that they obtain theoretically *optimal* rates.
- Though clipping speeds up SGD, it does not close the gap between SGD and ADAM. In section 4, we motivated the a novel *adaptive*-threshold coordinate-wise clipping algorithm and in section 5 experimentally show that it outperforms Adam on BERT training tasks.

## 1.1 Related work

**Adaptive step sizes.** Adaptive step sizes during optimization have long been studied [3, 21]. More recently, Duchi et al. [10] developed the Adagrad algorithm that benefits from the sparsity in stochastic gradients. Inspired by Adagrad, several adaptive methods have been proposed in the deep learning community [14, 28]. Recently, there has been a surge in interest to study the theoretical properties of these adaptive gradient methods due to [23], which pointed out the non-convergence of Adam and proposed an alternative algorithm, AMSGrad. Since then, many works studied different interesting aspects of adaptive methods, see [1, 6, 13, 16–18, 27, 30, 32–36]. Another direction of related work is normalized gradient descent, which has been studied for quasi-convex and non-convex settings [12, 15]. In contrast to our work, these prior works assume standard noise distributions that might not be applicable to key modern applications such as attention models, which exhibit heavy-tailed noise. Furthermore, convergence rates of adaptive methods are mostly worse than SGD.

**Noise in neural network.** There has been little study of the actual stochastic gradient noise distributions in neural network training. To our knowledge, [19, 25, 26] start the topic and observe heavy tailed noise in network training. Our work differs in two important ways: *First*, we treat the noise as a high dimensional vector, while [25] treat deviations in each coordinate as scaler noises to estimate tail index. Hence, we observe that the example given in [25] is well-concentrated when viewed as a random vector. This is also confirmed by [20]. More experimental comparisons are in Appendix H. *Second*, we focus on convergence of optimization algorithm, the previously mentioned works focus on Langevin dynamics and escaping saddle points. The convergence rate given in [19] is for global Holder-continuous functions, which restricts the function variations and excludes examples like quadratic functions. Our analysis instead provides the first convergence rates under the standard L-smoothness setting. Further, [11] studies accelerated first order methods under less concentrated noise, however, there "heavy-tailedness" refers to non-sub-Gaussianity.

## 2 Heavy-tailed noise in stochastic gradients

To gain intuition about the difference between SGD and adaptive methods, we start our discussion with the study of noise distributions of stochastic gradient that arise during neural network training. In particular, we focus on noise distributions while training two popular deep learning models — BERT and ResNet. Note that BERT and ResNet are typically trained with Adam and SGD (with momentum) respectively and can thus, provide insights about difference between these optimizers.

We first investigate the distribution of the gradient noise norm $\|g - \nabla f(x)\|$ in the aforementioned neural network models, where $g$ is the stochastic gradient computed from a minibatch sample. In particular, we fix the model at initialization without doing any updates. We then iterate through the

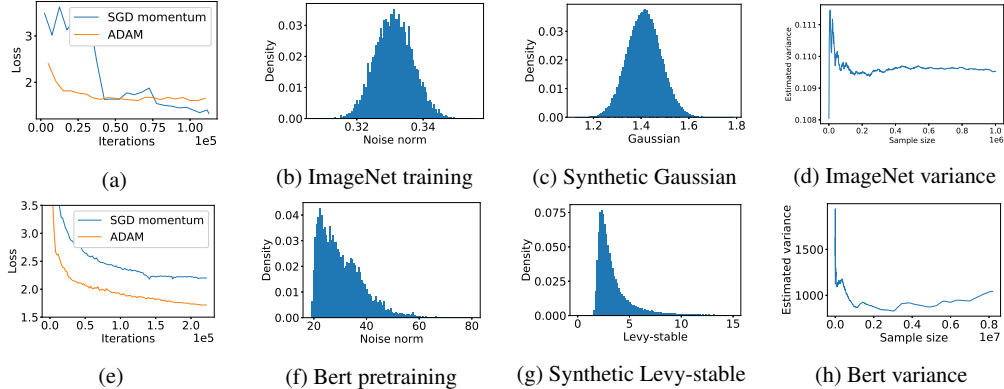

Figure 1: (a) Validation loss for ResNet50 trained on ImageNet. SGD momentum outperforms Adam. (b) Histogram of sampled gradient noise for ResNet50 on Imagenet dataset. (c) Histogram of samples from a sum of squared Gaussians. (d) Estimated variance of the stochastic gradient for Resnet50. (e)Validation loss for BERT pretraining. Although hyperparameters for SGD are finetuned, a large performance gap is still observed between SGD and Adam. (f) Histogram of sampled gradient nosie for BERT on Wikipedia+Books dataset. (g) Histogram of samples from a sum of squared $\alpha$-stable random variables. (h) Estimated variance of the stochastic gradient for BERT model.

dataset to compute the noise norm for each minibatch. Figure 1 (b) and (f) show these distributions for ResNet50 on ImageNet and BERT on the Wikipedia and books dataset at model initialization respectively. For comparison, we plot distributions of a normalized sum of squared Gaussians, a well-concentrated distribution, and a Levy-$\alpha$-stable distribution, a heavy-tailed distribution, in Figure 1 (c) and (g) respectively. We observe that the noise distribution for BERT appears heavy-tailed, while that of ResNet50 is well-concentrated. Results for noise distributions at other stages of training are displayed in Figure 2.

To support this observation, in Figure 1 (d) and (h) we further show the empirical variance of stochastic gradients with respect to the sample size used in the estimation. The results highlight that while the corresponding estimator converges for Imagenet, the empirical variance does not converge in BERT training even as the sample size approaches $10^7$.

From the obeservation that the noise can be heavy-tailed, we hypothesize that this is one major aspect that determines the performance of SGD and adaptive methods. In the rest of the paper, we argue and provide evidence that adaptive methods can be faster than SGD in scenarios where heavy-tailed noise distributions arise. More experiment details can be found in Section 5.

## 3 Convergence of gradient methods under heavy-tailed noise

In this section we study the performance of SGD and adaptive methods under heavy-tailed noise. More precisely, we analyze algorithms of the following form

$$x_{k+1} = x_k - \eta_k g_k, \tag{1}$$

where $x_k$ represent the current parameters, $\eta_k$ is the step size and $g_k$ is the stochastic (mini-batch) gradient evaluated at $x_k$. We show that if the stochasticity in the gradient $g_k$ is heavy-tailed, it is critical for the step sizes to be **adaptive** i.e. $\eta_k$ must depend on the observed gradients. We propose to use one such algorithm GClip and prove that it obtains optimal convergence rates.

**Heavy-tailed noise.** Neural network training can be seen as minimizing a differentiable stochastic function $f(x) = \mathbb{E}_\xi[f(x, \xi)]$, where $f : \mathbb{R}^d \to \mathbb{R}$ can be potentially nonconvex and $\xi$ represent the mini-batches. At each iteration, we assume access to an *unbiased* stochastic gradient $\mathbb{E}[g(x)] = \nabla f(x, \xi)$ corresponding to the parameters $x$, mini-batch $\xi$. We also need to bound how much noise is present in our stochastic gradients. In lieu of the usual bounded variance assumption, we use

**Assumption 1 (Bounded $\alpha-$moment).** *There exists positive real numbers $\alpha \in (1, 2]$ and $G > 0$ such that for all $x$, $\mathbb{E}[\|g(x) - \nabla f(x)\|^\alpha] \leq \sigma^\alpha$. We say noise is **heavy-tailed** if $\alpha < 2$.*

The above assumption with $\alpha = 2$ corresponds to the standard variance bound, but in general is weaker. It is indeed possible (e.g. Pareto or $\alpha$-stable Levy random variables) for the variance of $g(x)$

Table 1: Error bounds ($f(x) - f^*$ for convex functions, $\|\nabla f(x)\|$ for nonconvex functions) after $k$ iterations: Define $\alpha$-moment as $\mathbb{E}[\|g(x) - \nabla f(x)\|^\alpha] \le \sigma^\alpha$ (Assump 1) in the smooth nonconvex case and $\mathbb{E}[\|g(x)\|^\alpha] \le G^\alpha$ (Assump 4) in the strongly case. In the standard setting ($\alpha = 2$), GClip recovers the optimal rates. For heavy-tailed noise ($\alpha \in (1,2)$), GClip converges both for convex (Thm 4) and non-convex functions (Thm 2). We also show matching lower-bounds for all $\alpha \in (1,2]$ proving the optimality of clipping methods (Thm 5).

|  | Strongly Convex Function | | Non-Convex Function | |
|---|---|---|---|---|
|  | Heavy-tailed noise ($\alpha \in (1,2)$) | Standard noise ($\alpha \ge 2$) | Heavy-tailed noise ($\alpha \in (1,2)$) | Standard noise ($\alpha \ge 2$) |
| SGD | N/A | $\mathcal{O}(k^{-1})$ | N/A | $\mathcal{O}(k^{-\frac{1}{4}})$ |
| GClip | $\mathcal{O}(k^{\frac{-(\alpha-1)}{\alpha}})$ | $\mathcal{O}(k^{-1})$ | $\mathcal{O}(k^{\frac{-(\alpha-1)}{3\alpha-2}})$ | $\mathcal{O}(k^{-\frac{1}{4}})$ |
| LowerBound | $\Omega(k^{\frac{-(\alpha-1)}{\alpha}})$ | $\Omega(k^{-1})$ | $\Omega(k^{\frac{-(\alpha-1)}{3\alpha-2}})$ | $\Omega(k^{-\frac{1}{4}})$ |

to be unbounded, while simultaneously satisfying assumption 1 for $\alpha < 2$. One should note that even if the variance may not actually be infinite in practice, it might be too large to be practically useful. All our analyses and insights carry over to this setting as well.

The possibility that the variance is unbounded has a profound impact on the optimization process.

**Remark 1** (Nonconvergence of SGD). *Consider the function $f(x) = x^2/2$ with noise satisfying $\mathbb{E}[\|g(x) - \nabla f(x)\|^\alpha] = \sigma^\alpha$ for $\alpha < 2$, and $\mathbb{E}[\|g(x) - \nabla f(x)\|^2] = \infty$. Then, for any positive constants $\eta_k$ that do not depend on $g_k$, we have that $\mathbb{E}[\|\nabla f(x_k)\|^2] = \infty$.*

*Proof.* we denote the stochastic gradient $g_k := g(x_k) = \nabla f(x_k) + \xi_k = x_k + \xi_k$, where $\xi_k \in \mathbb{R}^d$ is a random variable with $\mathbb{E}\|\xi\|^2 = \infty, \mathbb{E}\|\xi\|^\alpha = \sigma^\alpha, \mathbb{E}[\xi] = \vec{0}$. Then, $\mathbb{E}[\|\nabla f(x_{k+1})\|^2] = \mathbb{E}[\|x_{k+1}\|^2] = \mathbb{E}\|x_k - \eta_k g_k\|^2 = \mathbb{E}\|x_k - \eta_k(x_k + \xi)\|^2 = \mathbb{E}\|(1 - \eta_k)x_k - \eta_k\xi\|^2 = \mathbb{E}\|(1 - \eta_k)x_k\|^2 - 2(1 - \eta_k)\eta_k x_k^\top \mathbb{E}[\xi] + \eta_k^2 \mathbb{E}\|\xi\|^2 \ge \infty$. Note that this holds for *any* fixed $\eta_k > 0$ even if allowed to depend on the statistics of the noise distribution (such as $\sigma$ or $\alpha$). $\square$

The issue is that SGD is easily influenced by a single-stochastic gradient, which could be very large and incorrect. A simple strategy to circumvent this issue is to use a biased *clipped* stochastic gradient estimator. This allows us to circumvent the problem of unbounded variance and ensures optimal convergence rates even under heavy-tailed noise. Our results are summarized in Table 1, and all proofs are relegated to the Appendices.

### 3.1 Convergence of Clipped Methods

A simple clipping strategy is to globally clip the norm of the update to threshold $\tau_k$:

$$x_{k+1} = x_k - \eta_k \min\left\{\frac{\tau_k}{\|g_k\|}, 1\right\} g_k, \ \tau_k \in \mathbb{R}_{\ge 0} \tag{GClip}$$

We refer to this strategy as GClip (Global Clip), as opposed to coordinate-wise clipping which we discuss later. We first state the rates for smooth non-convex functions.

**Theorem 2** (**Non-convex convergence**). *Suppose that $f$ is $L$-smooth and that the stochastic gradients satisfy Assumption 1 for $\alpha \in (1,2]$. Let $\{x_k\}$ be the iterates of GClip with parameters $\eta_k = \eta = \min\{\frac{1}{4L}, \frac{\sigma^\alpha}{L\tau^\alpha}, \frac{1}{24L\tau}\}$ and $\tau_k = \tau = \max\{2, 48^{1/(\alpha-1)}\sigma^{\alpha/(\alpha-1)}, 8\sigma, \left(\frac{f_0}{\sigma^2 K}\right)^{\frac{\alpha}{3\alpha-2}} / L^{\frac{2\alpha-2}{3\alpha-2}}, \}$. Then for $F_0 := f(x_0) - f^*$,*

$$\frac{1}{K}\sum_{k=1}^{K} \mathbb{E}[\min\{\|\nabla f(x_k)\|, \|\nabla f(x_k)\|^2\}] = \mathcal{O}(K^{\frac{-2\alpha+2}{3\alpha-1}}), .$$

**Remark 3.** *When $\|\nabla f(x_k)\| \le \epsilon \ll 1$, $\|\nabla f(x_k)\|^2 \ll \|\nabla f(x_k)\|$. Hence the minimum term on the left hand side of the inequality above is $\|\nabla f(x_k)\|^2$. The right hand side is easily observed to be $\mathcal{O}(K^{-\frac{2(\alpha-1)}{3\alpha-2}})$. Together, this implies a convergence rate of $\mathbb{E}\|\nabla f(x)\| \le \mathcal{O}(K^{-\frac{(\alpha-1)}{3\alpha-2}})$.*

We prove improved rates of convergence for non-smooth *strongly-convex* functions in a bounded domain. Due to limited space, we relegate the definitions and assumptions to Appendix A.

**Theorem 4 (Strongly-convex convergence).** *Suppose that the stochastic gradients satisfy Assumption 4 for $\alpha \in (1,2]$. Let $\{x_k\}$ be the iterates of projected GClip (proj-GClip) with clipping parameter $\tau_k = Gk^{\alpha-1}$ and steps-size $\eta_k = \frac{4}{\mu(k+1)}$. Define the output to be a $k$-weighted combination of the iterates: $\bar{x}_k = \sum_{j=1}^{k} jx_{j-1}/(\sum_{j=1}^{k} j)$ . Then the output $\bar{x}_k$ satisfies:*

$$\mathbb{E}[f(\bar{x}_k)] - f(x^\star) \leq \frac{16G^2}{\mu(k+1)^{2(\alpha-1)/\alpha}} \; .$$

The rates of convergence for the strongly convex and non-convex cases in Theorem 4 and Theorem 2 exactly match those of the usual SGD rates ($\mathcal{O}(1/\sqrt{k})$ for convex and $\mathcal{O}(k^{-\frac{1}{4}})$ for non-convex) when $\alpha = 2$ and gracefully degrade for $\alpha \in (1,2]$. As we will next show, both the strongly convex rates and non-convex rates of GClip are in fact optimal for every $\alpha \in (1,2]$.

### 3.2 Theoretic lower bounds

We prove that the rates obtained with GClip are optimal up to constants. First, we show a strong lower-bound for the class of convex functions with stochastic gradients satisfying $\mathbb{E}[|g(x)|^\alpha] \leq 1$. This matches the upper bounds of Theorems 4 and 8 for strongly-convex functions, showing that the simple clipping mechanism of GClip is (up to constants) information theoretically optimal, providing a strong justification for its use.

**Theorem 5.** *For any $\alpha \in (1,2]$ and any (possibly randomized) algorithm $\mathcal{A}$, there exists a problem $f$ which is 1-strongly convex and 1-smooth ($\mu = 1$ and $L = 1$), and stochastic gradients which satisfy Assumptions 4 with $G \leq 1$ such that the output $x_k$ of the algorithm $\mathcal{A}$ after processing $k$ stochastic gradients has an error*

$$\mathbb{E}[f(x_k)] - f(x^\star) \geq \Omega\left(\frac{1}{k^{2(\alpha-1)/\alpha}}\right) .$$

Next, we examine non-convex functions.

**Theorem 6.** *Given any $\alpha \in (1,2]$, smoothness constant $L$, and (possibly randomized) algorithm $\mathcal{A}$, there exists a constant $c_1$ and an $L$-smooth function $f$ with stochastic gradients satisfying Assumption 1 for any given $\sigma \geq c_1\sqrt{(f(0) - f^\star)L}$ such that the output $x_k$ of the algorithm $\mathcal{A}$ after processing $k$ stochastic gradients has an error*

$$\mathbb{E}[\|\nabla f(x_k)\|] \geq \Omega\left(\frac{1}{k^{(\alpha-1)/(3\alpha-2)}}\right) .$$

Theorem 6, proven in Appendix G, extends the recent work of [2, Theorem 1] to heavy-tailed noise. Here, the lower-bound matches the upper-bound in Theorem 2 up to constants, proving its optimality.

## 4 Faster Optimization with Adaptive Coordinate-wise Clipping

The previous section showed that adaptive step sizes (which depend on the gradients) are essential for convergence under heavy-tailed noise, and also showed that GClip provides the optimal rates. There are of course other adaptive methods such as Adam which employs not only the current gradients but also *all* past gradients to adaptively set *coordinate-wise* step-sizes. In this section, we study why coordinate-wise clipping may yield even faster convergence than GClip, and show how to modify GClip to design an Adaptive Coordinate-wise Clipping algorithm (ACClip).

### 4.1 Coordinate-wise clipping

The first technique we use is applying coordinate-wise clipping instead of global clipping. We had previously assumed a global bound on the $\alpha$-moment of the *norm* (or variance) of the stochastic gradient is bounded by $\sigma$. However, $\sigma$ might be hiding some dimension dependence $d$. We show a more fine-grained model of the noise in order to tease out this dependence.

**Assumption 2 (Coordinate-wise $\alpha$ moment).** *Denote $\{g_i(x)\}$ to be the coordinate-wise stochastic gradients for $i \in [d]$. We assume there exist constants $\{B_i\} \geq 0$ and $\alpha \in (1,2]$ such that $\mathbb{E}[|g_i(x)|^\alpha] \leq B_i^\alpha$ .*

For the sake of convenience, we denote $B = [B_1; B_2; \cdots ; B_d] \in \mathbb{R}^d$, $\|B\|_a = (\sum B_i^a)^{1/a}$. Under this more refined assumption, we can show the following corollary:

**Corollary 7 (GClip under coordinate-wise noise).** *Suppose we run GClip under Assumption 2 to obtain the sequence $\{x_k\}$. If $f$ is $\mu$-strongly convex, with appropriate step-sizes and averaging, the output $\bar{x}_k$ satisfies*

$$\mathbb{E}[f(\bar{x}_k)] - f(x^\star) \leq \frac{16d\|B\|_\alpha^2}{\mu(k+1)^{2(\alpha-1)/\alpha}} \; .$$

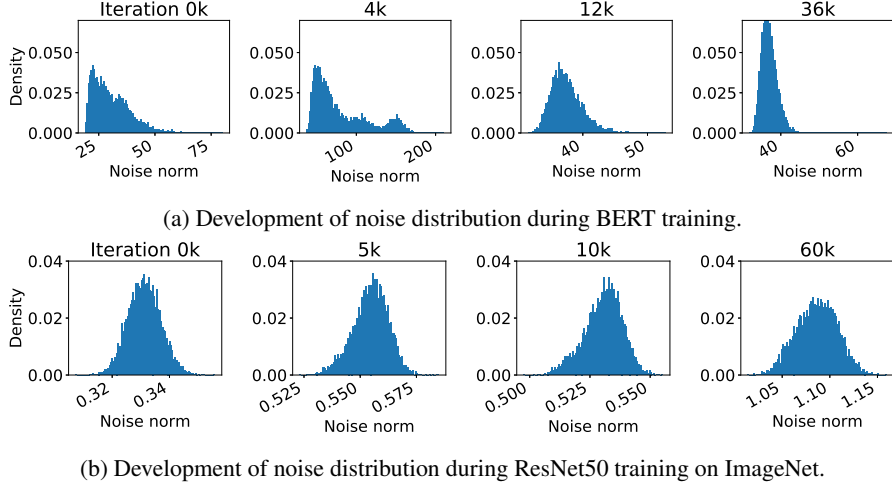

(a) Development of noise distribution during BERT training.

(b) Development of noise distribution during ResNet50 training on ImageNet.

Figure 2: The distribution of gradient noise is non-stationary during BERT training, while it remains almost unchanged for ResNet training on ImageNet.

Thus, the convergence of GClip can have a strong dependence on $d$, which for large-scale problems might be problematic. We show next that using coordinate-wise clipping removes this dependency:

$$x_{k+1} = x_k - \eta_k \min\left\{\frac{\tau_k}{|g_k|}, 1\right\} g_k , \ \tau_k \in \mathbb{R}^d_{\geq 0} . \tag{CClip}$$

**Theorem 8** (**CClip under coordinate-wise noise**). *Suppose we run CClip under the Assumption of 2 with $\tau_k = Bk^{\alpha-1}$ to obtain the sequence $\{x_k\}$. Then, if $f$ is $\mu$-strongly convex, with appropriate step-sizes and averaging, the output $\bar{x}_k$ satisfies*

$$\mathbb{E}[f(\bar{x}_k)] - f(x^\star) \leq \frac{16\|B\|_2^2}{\mu(k+1)^{2(\alpha-1)/\alpha}} .$$

Note that $\|B\|_2 \leq \|B\|_\alpha$. CClip has a worst-case convergence independent of $d$ under the coordinate-wise noise model. Similar comparison between GClip and CClip can be done for non-convex conditions too, but we skip for conciseness. Though we only compare upper-bounds here, when the noise across coordinates is independent the upper bounds may be tight (see Lemma 12).

### 4.2 Online moment estimation

We now present the second technique that is motivated by our observation in Figure 2. There, the distribution of gradient noise at the beginning of different epochs is shown during training for BERT with Wikipedia (top) as well as ResNet with ImageNet (bottom). The result highlights that the noise distribution is not only heavy-tailed, but also non-stationary during BERT training and becomes increasingly more concentrated. In contrast, for the ResNet model the noise distribution remains mostly unchanged.

Since the scale of the noise changes drastically during training for BERT model and our theoretical analysis suggest that we should clip proportional to the noise level, we propose to use an exponential moving average estimator to estimate the moment and clip the gradient accordingly (line 4,5 of Alg 1). This, combined with the momentum term leads to our proposed ACClip algorithm in Algorithm 1. On a high level, the algorithm applies clipping to the momentum term, where the clipping threshold is proportional to the estimated moment using an exponential moving average. From our experiment, we found the conservative choice of $\alpha = 1$ leads to the best performance.

## 5 Experiments

In this section, we first verify the effect of coordinate-wise clipping and moment estimation introduced in Section 4. We then perform extensive evaluations of ACClip on BERT pre-training and fine-tuning tasks and demonstrate its advantage over Adam in Section 5.2. For completeness, an experiment on ImageNet is included in Appendix I. Finally, we start with a few more experiments on the noise distribution in neural network training.

**Algorithm 1** ACClip

1: $x, m_k \leftarrow x_0, 0$
2: **for** $k = 1, \cdot, T$ **do**
3: $\quad m_k \leftarrow \beta_1 m_{k-1} + (1 - \beta_1) g_k$
4: $\quad \tau_k^\alpha \leftarrow \beta_2 \tau_{k-1}^\alpha + (1 - \beta_2) |g_k|^\alpha$
5: $\quad \hat{g}_k \leftarrow \min\left\{\frac{\tau_k}{|m_k| + \epsilon}, 1\right\} m_k$
6: $\quad x_k \leftarrow x_{k-1} - \eta_k \hat{g}_k$
7: **end forreturn** $x_K$, where random variable $K$ is supported on $\{1, \cdots, T\}$.

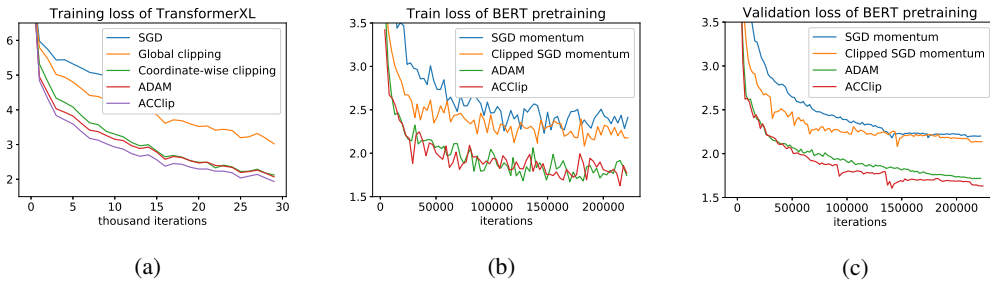

(a)               (b)              (c)

Figure 3: (a) Performance of different algorithms for training a toy transformer-XL model described in Section 4. (b) Train and (c) validation loss for $\text{BERT}_{base}$ pretraining with the sequence length of 128. While there remains a large gap between non-adaptive methods and adaptive methods, clipped SGD momentum achieves faster convergence compared to standard SGD momentum. The proposed algorithm for adaptive coordinate-wise clipping (ACClip) achieves a lower loss than Adam.

## 5.1 From GClip to ACClip

In this section we instantiate the argument in Section 4 with a set of experiments. As seen in Figure 3b, global clipping improves the vanilla SGD algorithm but is still far from the ADAM baseline. We apply two techniques (coordinate-wise clipping and online moment estimation) onto the clipped SGD algorithm analyzed in Section 3. We use a set of experiments on Transformer-XL training to demonstrate the effect of each technique.

**Experiment setup**    We train a 6-layer Transformer-XL model[8] on PTB dataset as a proof of concept. Our main experiments will be on BERT pretraining and finetuning described in the next subsection 5.2. We adapt the author's github repo[1], and replace the number of layers of the base model by 6. We then select the PTB data as input and set the maximum target length to be 128. The results are shown in Figure 3b.

**Observations**    From Figure 3b, we can tell that global clipping (orange curve) indeed speeds up vanilla SGD but is still much worse compared to the ADAM baseline provided by the code base. After replacing global clipping with coordinate-wise clipping, we see that the performance is already comparable to the ADAM baseline. Finally, after using the moment estimation to determine the clipping threshold, we are able to achieve faster convergence than ADAM.

## 5.2 Performance of ACClip for BERT pre-training and fine-tuning

We now evaluate the empirical performance of our proposed ACClip algorithm on BERT pre-training as well fine-tuning using the SQUAD v1.1 dataset. As a baseline, we use Adam optimizer and the same training setup as in the BERT paper [9]. For ACClip, we set $\tau = 1$, *learning rate* = 1e-4, $\beta_1 = 0.9$, $\beta_2 = 0.99$, $\epsilon$ = 1e-5 and *weight decay* = 1e-5. We compare both setups on BERT models of three different sizes, $\text{BERT}_{base}$ with 6 and 12 layers as well as $\text{BERT}_{large}$ with 24 layers.

Figure 3b and 3c shows the loss for pretraining $\text{BERT}_{base}$ using SGD with momentum, GClip, Adam and ACClip. The learning rates and hyperparameters for each method have been extensively

Table 2: **BERT pretraining: Adam vs ACClip.** Compared to Adam, the proposed ACClip algorithm achieves better evaluation loss and Masked LM accuracy for all model sizes.

| | BERT Base 6 layers | | BERT Base 12 layers | | BERT Large 24 layers | |
|---|---|---|---|---|---|---|
| | Val. loss | Accuracy | Val. loss | Accuracy | Val. loss | Accuracy |
| Adam | 1.907 | 63.45 | 1.718 | 66.44 | 1.432 | 70.56 |
| ACClip | **1.877** | **63.85** | **1.615** | **67.16** | **1.413** | **70.97** |

Table 3: **SQUAD v1.1 dev set: Adam vs ACClip**. The mean and standard deviation of F1 and exact match score for 5 runs. The first row contains results reported from the original BERT paper, which are obtained by picking the best ones out of 10 repeated experiments.

| | BERT Base 6 layers | | BERT Base 12 layers | | BERT Large 24 layers | |
|---|---|---|---|---|---|---|
| | EM | F1 | EM | F1 | EM | F1 |
| Adam (Devlin et al., 2018) | | | 80.8 | 88.5 | 84.1 | 90.9 |
| Adam | $76.85 \pm 0.34$ | $84.79 \pm 0.33$ | $81.42 \pm 0.16$ | $88.61 \pm 0.11$ | $83.94 \pm 0.19$ | $90.87 \pm 0.12$ |
| ACClip | $\mathbf{78.07 \pm 0.24}$ | $\mathbf{85.87 \pm 0.13}$ | $\mathbf{81.62 \pm 0.18}$ | $\mathbf{88.82 \pm 0.10}$ | $\mathbf{84.93 \pm 0.29}$ | $\mathbf{91.40 \pm 0.15}$ |

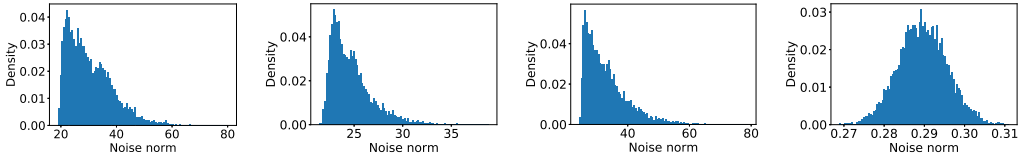

(a) Attention + Wikipedia  (b) Attention + Gaussian  (c) Resnet + Wikipedia  (d) Resnet + Gaussian.

Figure 4: Distribution of gradient noise norm in Attention and ResNet models on two data sources: Wikipedia and synthetic Gaussian. The heavy-tailed noise pattern results from the interaction of both model architecture as well as data distribution.

tuned to provide best performance on validation set. However, even after extensive tuning, there remains a large gap between (clipped) SGD momentum and adaptive methods. Furthermore, clipped SGD achieves faster convergence as well as lower final loss compared to standard SGD. Lastly, the proposed optimizer ACClip achieves a lower loss than the Adam. Table 2 further shows that ACClip achieves lower loss and higher masked-LM accuracy for all model sizes.

Next, we evaluate ACClip on the SQUAD v1.1 fine-tuning task. We again follow the procedure outlined in [9] and present the results on the Dev set in Table 3. Both for F1 as well as for exact match, the proposed algorithm outperforms Adam on all model sizes. The experimental results on BERT pretraining and fine-tuning indicate the effectiveness of the proposed algorithm.

### 5.3 Noise Patterns in BERT and ImageNet Training

In our initial analysis in Figure 1, we observe that training an attention model on Wikipedia leads to heavy-tailed noise whereas training a ResNet on ImageNet data leads to well-concentrated noise. Here, we aim to disentangle the effect that model architecture and training data have on the shape of gradient noise. To this end, we measure the distribution of the gradient noise norm in an Attention and a ResNet model on both Wikipedia and synthetic Gaussian data. We used BERT$_{base}$ as the Attention model, and the ResNet is constructed by removing the self-attention modules within the transformer blocks. Gaussian synthetic data is generated by replacing the token embedding layer with normalized Gaussian input. The resulting noise histograms are shown in Figure 4. The figure shows that the Attention model leads to heavy-tailed noise independently of input data. For the ResNet model, we observe that Gaussian input leads to Gaussian noise, whereas Wikipedia data leads to be heavy-tailed noise. We thus conclude that the heavy-tailed noise pattern results from both the model architecture as well as the data distribution.

## 6 Discussion

One immediate extension from this work is to view RMSProp as a clipping algorithm and prove its convergence under shifting noise. The update for RMSProp and ACClip with $\beta_1 = 0$ can be written

with effective step-sizes $h_{\text{rms}}$ and $h_{\text{clip}}$ respectively as below:

$$x_{k+1} = x_k - \frac{\alpha}{\epsilon + \sqrt{\beta_2 v_k + (1-\beta_2)|g_k|^2}} g_k =: x_k - h_{\text{Adam}} g_k \text{ , and}$$

$$x_{k+1} = \quad x_k - \eta_k \min\left\{\frac{\tau_k}{|g_k|}, 1\right\} g_k \quad =: x_k - h_{\text{clip}} g_k.$$

Given any set of parameters for RMSProp, if we set the parameters for ACClip as

$$\eta_k = \frac{2\alpha}{\epsilon + \sqrt{\beta_2 v_k}} \quad \text{and} \quad \tau_k = \frac{\epsilon + \sqrt{\beta_2 v_k}}{\sqrt{1-\beta_2}},$$

then $\frac{1}{2} h_{\text{clip}} \leq h_{\text{Adam}} \leq 2h_{\text{clip}}$. Thus, RMSProp can be seen as ACClip where $\tau_k$ is set using $\sqrt{v_k}$, which estimates $\mathbb{E}[|g_k|^2]^{1/2}$, and a correspondingly decreasing step-size. An analysis of RMSprop (and Adam) by viewing them as adaptive clipping methods is a great direction for future work.

In summary, our work theoretically and empirically ties the advantage of adaptive methods over SGD to the heavy-tailed nature of gradient noise. A careful analysis of the noise and its impact yielded two insights: that clipping is an excellent strategy to deal with heavy-tailed noise, and that the ACClip yields state of the art performance for training attention models. Our results add to a growing body of work which demonstrate the importance of the structure of the noise in understanding neural network training. We believe additional such investigations into the source of the heavy tailed-ness, as well as a characterization of the noise can lead to further insights with significant impact on practice.

## Broader impact

We study convergence rates of gradient methods under a more relaxed noise condition. The result under this setting reaches conclusions that are closer to practice compared to results under the standard setting. Hence, our work provides one way to bridge the theory-practice gap and can facilitate more future works in this direction.

## Acknowledgement

This work was done when Jingzhao and Praneeth were interns at Google. Suvrit and Jingzhao also acknowledge support from NSF CAREER grant Number 1846088 and an An Amazon Research Award(ARA).

## Footnotes

[1]https://github.com/kimiyoung/transformer-xl/tree/master/pytorch

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
