[Supplementary Material]

# A Additional definitions and assumptions

Here we describe some of the formal assumptions which were previously skipped.

## A.1 Assumptions in the nonconvex setting

We define the standard notion of smoothness.

**Assumption 3** (*L*-**smoothness**). *$f$ is $L$-smooth, i.e. there exist positive constants $L$ such that $\forall x, y$,*
$f(y) \leq f(x) + \langle \nabla f(x), y - x \rangle + \frac{L}{2} \|y - x\|^2$.

Note that we only need the smoothness assumption for non-convex functions.

## A.2 Assumptions in the strongly convex setting

For strongly-convex optimization, instead of bounding the noise, we assume that the stochastic oracle has bounded moment.

**Assumption 4** (**bounded $\alpha$ moment**). *There exists positive real numbers $\alpha \in (1, 2]$ and $G > 0$ such that for all $x$, $\mathbb{E}[\|g(x)\|^\alpha] \leq G^\alpha$.*

Note that the above assumption implies a uniform bound on gradient norm. Such bound is necessary for nonsmooth strongly convex problems, as one can no longer factor out the gradient norm using the smoothness assumption. See for example, [22].

**Assumption 5** ($\mu$-**strong-convexity**). *$f$ is $\mu$-strongly convex, if there exist positive constants $\mu$ such that $\forall x, y$,*

$$f(y) \geq f(x) + \langle \nabla f(x), y - x \rangle + \frac{\mu}{2} \|y - x\|^2$$

The strong convexity assumption and the bounded gradient assumption implies that the domain is bounded, which we state explicitly below,

**Assumption 6** (**bounded domain**). *We look for a solution $x$ within a bounded convex set $\mathcal{X}$.*

We didn't upper bound the domain diameter as it is not used explicitly in the proof. To ensure all updates are within a domain, we use the projected version of (GClip) defined as follows:

$$x_{k+1} = \text{proj}_\mathcal{X}\{x_k - \eta_k \min\{\tfrac{\tau_k}{\|g_k\|}, 1\}g_k \,, \; \tau_k \in \mathbb{R}_{\geq 0}]\} \qquad \text{(proj-GClip)}$$

The projection operator $x = \text{proj}_\mathcal{X}(y)$ finds the point $x \in \mathcal{X}$ that has the least distance to $y$.

# B Effect of global clipping on variance and bias

We focus on (GClip) under stochastic gradients which satisfy Assumption 1.

**Lemma 9.** *For any $g(x)$ suppose that assumption 1 holds with $\alpha \in (1, 2]$. If $\mathbb{E}[\|g(x)\|^\alpha] \leq G^\alpha$, then the estimator $\hat{g} := \min\{\tfrac{\tau_k}{\|g_k\|}, 1\}g_k$ from (GClip) with clipping parameter $\tau \geq 0$ satisfies:*

$$\mathbb{E}[\|\hat{g}(x)\|^2] \leq G^\alpha \tau^{2-\alpha} \text{ and } \|\mathbb{E}[\hat{g}(x)] - \nabla f(x)\|^2 \leq G^{2\alpha} \tau^{-2(\alpha-1)}.$$

*Proof.* First, we bound the variance.
$$\mathbb{E}[\|\hat{g}(x)\|^2] = \mathbb{E}[\|\hat{g}(x)\|^\alpha \|\hat{g}(x)\|^{2-\alpha}]$$
By the fact that $\hat{g}(x) \leq \tau$, we get
$$\mathbb{E}[\|\hat{g}(x)\|^2] = \mathbb{E}[\|\hat{g}(x)\|^\alpha \tau^{2-\alpha}] \leq G^\alpha \tau^{2-\alpha}.$$
Next, we bound the bias,
$$\begin{aligned}
\|\mathbb{E}[\hat{g}(x)] - \nabla f(x)\| &= \|\mathbb{E}[\hat{g}(x) - g(x)]\| \\
&\leq \mathbb{E}[\|\hat{g}(x) - g(x)\|] = \mathbb{E}[\|\hat{g}(x) - g(x)\| \mathbb{1}_{\{|g(x)|\geq\tau\}}] \\
&\leq \mathbb{E}[\|g(x)\| \mathbb{1}_{\{|g(x)|\geq\tau\}}] \\
&\leq \mathbb{E}[\|g(x)\|^\alpha \mathbb{1}_{\{|g(x)|\geq\tau\}}] / \tau^{\alpha-1}.
\end{aligned}$$

The first inequality follows by Jenson's inequality. The second inequality follows by definition of $\hat{g}$. The third inequality follows by $\|g(x)\|^{\alpha}\mathbb{1}_{\{|g(x)|\geq\tau\}} \geq \|g(x)\|\tau^{\alpha-1}\mathbb{1}_{\{|g(x)|\geq\tau\}}$. $\qquad\square$

As we increase the clipping parameter $\tau$, note that the variance (the first term in Lemma 9) increases while the bias (which is the second term) decreases. This way, we can carefully trade-off the variance of our estimator against its bias, thereby ensuring convergence of the algorithm.

## C  Non-convex Rates (Proof of Theorem 2)

The lemma in the previous section can be readily used in the nonsmooth strongly convex setting. However, we need a variant of Lemma 9 in the smooth case.

**Lemma 10.** *For any $g(x)$ suppose that assumption 1 holds with $\alpha \in (1,2]$. If $\|\nabla f(x)\| \leq \tau/2$, then the estimator $\hat{g} := \min\{1, \tau/\|g_k\|\}g_k$ from (GClip) with global clipping parameter $\tau \geq 0$ satisfies:*

$$\mathbb{E}\big[\|\hat{g}(x)\|^2\big] \leq 2\|\nabla f(x)\|^2 + 4\sigma^{\alpha}\tau^{2-\alpha} \ and \ \|\mathbb{E}[\hat{g}(x)] - \nabla f(x)\|^2 \leq 4\sigma^{2\alpha}\tau^{-2(\alpha-1)}\ .$$

*Proof.* First, we bound the variance.

$$
\begin{aligned}
\mathbb{E}[\|\hat{g}(x)\|^2] &\leq \mathbb{E}[2\|\nabla f(x)\|^2 + 2\|\nabla f(x) - \hat{g}(x)\|^2] \\
&= \mathbb{E}[2\|\nabla f(x)\|^2 + 2\|\nabla f(x) - \hat{g}(x)\|^{\alpha}\|\nabla f(x) - \hat{g}(x)\|^{2-\alpha}] \\
&\leq \mathbb{E}[2\|\nabla f(x)\|^2 + 2\|\nabla f(x) - \hat{g}(x)\|^{\alpha}(2\tau)^{2-\alpha}] \\
&\leq 2\|\nabla f(x)\|^2 + 4\tau^{2-\alpha}\mathbb{E}[\|\nabla f(x) - g(x)\|^{\alpha}] \\
&\leq 2\|\nabla f(x)\|^2 + 4\tau^{2-\alpha}\sigma^{\alpha}
\end{aligned}
$$

The expectation is taken with respect to the randomness in noise. The second last inequality follows by the fact that $\|\nabla f(x) - \hat{g}(x)\| < 2\tau$.

Next, we bound the bias,

$$
\begin{aligned}
\|\mathbb{E}[\hat{g}(x)] - \nabla f(x)\| &= \|\mathbb{E}[\hat{g}(x) - g(x)]\| \\
&= \mathbb{E}[|\|g(x)\| - \tau|\mathbb{1}_{\{\|g(x)\|>\tau\}}] \\
&\leq \mathbb{E}[\|g(x) - \nabla f(x)\|\mathbb{1}_{\{\|g(x)\|>\tau\}}] \\
&\leq \mathbb{E}[\|g(x) - \nabla f(x)\|\mathbb{1}_{\{\|g(x)-\nabla f(x)\|>\tau/2\}}] \\
&\leq \mathbb{E}[\|g(x) - \nabla f(x)\|^{\alpha}](\tau/2)^{1-\alpha} \leq 2\sigma^{\alpha}\tau^{1-\alpha}
\end{aligned}
$$

The last line follows by

$$\|g(x) - \nabla f(x)\|\mathbb{1}_{\{\|g(x)-\nabla f(x)\|>\tau/2\}} \leq \frac{\|g(x) - \nabla f(x)\|^{\alpha}}{(\tau/2)^{\alpha-1}}\mathbb{1}_{\{\|g(x)-\nabla f(x)\|>\tau/2\}}.$$

$\qquad\square$

Next, we need a subprocedure at the end proof of Lemma 2 from [7].

**Lemma 11** (Lemma 2 in [7])**.** *For any vector $v \in \mathbb{R}^d$, $\langle v/\|v\|, \nabla f(x)\rangle \geq \frac{\|\nabla f(x)\|}{3} - \frac{8\|v - \nabla f(x)\|}{3}$.*

Finally, we are ready to show the proof.

*Proof.* At each iteration, we consider two cases, either $\|\nabla f(x_k)\| < \tau/2$ or $\|\nabla f(x_k)\| \geq \tau/2$.

**Case 1:** $\|\nabla f(x_k)\| < \tau/2$  For simplicity, we denote $\hat{g}_k = \min\{1, \tau/\|g_k\|\}g_k$ and the bias $b_k = \mathbb{E}[\hat{g}_k] - \nabla f(x_k)$. By Assumption 3, we have

$$
\begin{aligned}
f(x_k) &\leq f(x_{k-1}) + \langle \nabla f(x_{k-1}), -\eta_k\hat{g}_{k-1}\rangle + \frac{\eta_{k-1}^2 L}{2}\|\hat{g}_{k-1}\|^2 \\
&\leq f(x_{k-1}) - \eta_{k-1}\|\nabla f(x_{k-1})\|^2 - \eta_{k-1}\langle \nabla f(x_{k-1}), b_{k-1}\rangle + \frac{\eta_{k-1}^2 L}{2}\|\hat{g}_{k-1}\|^2 \\
&\leq f(x_{k-1}) - \eta_{k-1}\|\nabla f(x_{k-1})\|^2 + \frac{\eta_{k-1}}{2}\|\nabla f(x_{k-1})\|^2 + \frac{\eta_{k-1}}{2}\|b_{k-1}\|^2 + \frac{\eta_{k-1}^2 L}{2}\|\hat{g}_{k-1}\|^2\ .
\end{aligned}
$$

Here the last step used the Cauch Schwarz and the AM-GM inequalities. Then, taking expectation in both sides and using Lemma 10 gives

$$\mathbb{E}[f(x_k)|x_{k-1}] \leq f(x_{k-1}) - (\frac{\eta_{k-1}}{2} - \eta_{k-1}^2 L)\|\nabla f(x_{k-1})\|^2 + 2\eta_{k-1}^2 L\sigma^\alpha \tau^{2-\alpha} + \frac{2\eta_k \sigma^{2\alpha}}{\tau^{2\alpha-2}}$$

$$\leq f(x_{k-1}) - \frac{\eta_{k-1}}{8}\|\nabla f(x_{k-1})\|^2 + 2\eta_{k-1}^2 L\sigma^\alpha \tau^{2-\alpha} + \frac{2\eta_{k-1}\sigma^{2\alpha}}{\tau^{2\alpha-2}}.$$

In the last step we used $\{\eta_k = \eta \leq \frac{1}{4L}\}$.

**Case 2:** $\|\nabla f(x_k)\| > \tau/2$   Recall $\hat{g}_k = \min\{1, \tau/\|g_k\|\}g_k$ and parameter choices $\eta_k = \eta = \min\{\frac{1}{4L}, \frac{1}{L\tau^\alpha}, \frac{1}{24L\tau}\}$ and $\tau_k = \tau = \max\{2, 48^{1/(\alpha-1)}\sigma^{\alpha/(\alpha-1)}, 48\sigma, \sigma K^{\frac{1}{3\alpha-2}}\}$. We use $\nabla f$ as a shorthand for $\nabla f(x_k)$ and $p := \mathbb{E}[\mathbb{1}_{\{\|g_k\| \leq \tau\}}]$ :

$$\mathbb{E}[\langle \nabla f, g_k \rangle \mathbb{1}_{\{\|g_k\| \leq \tau\}}] \geq \mathbb{E}[(\|\nabla f\|^2 - \|\nabla f\|\|g_k - \nabla f\|)\mathbb{1}_{\{\|g_k\| \leq \tau\}}]$$
$$\geq \mathbb{E}[\|\nabla f\|^2 \mathbb{1}_{\{\|g_k\| \leq \tau\}} - \frac{1}{4}\|\nabla f\|^2 \mathbb{1}_{\{\|g_k\| \leq \tau, \|g_k - \nabla f\| \leq \tau/8\}} - \|\nabla f\|\|g_k - \nabla f\|\mathbb{1}_{\{\|g_k\| \leq \tau, \|g_k - \nabla f\| \geq \tau/4\}}]$$
$$\geq \frac{3p}{4}\|\nabla f\|^2 - \|\nabla f\|\mathbb{E}[\|g_k - \nabla f\|\mathbb{1}_{\{\|g_k - \nabla f\| \geq \tau/8\}}]$$
$$\geq \frac{3p}{4}\|\nabla f\|^2 - \|\nabla f\|\frac{\sigma^\alpha}{(\tau/8)^{\alpha-1}} \tag{2}$$

The first inequality uses

$$\langle \nabla f, g_k \rangle = \|\nabla f\|^2 + \langle \nabla f, g_k - \nabla f \rangle.$$

The second line follows by

$$\|\nabla f\| > \tau/2 \text{ and } \|g_k - \nabla f\| < \tau/8 \implies -\|\nabla f\|\|g_k - \nabla f\| \geq -\|\nabla f\|^2/4.$$

The last inequality follows by $\sigma^\alpha \geq \mathbb{E}[\|g_k - \nabla f\|^\alpha] \geq \mathbb{E}[\|g_k - \nabla f\|(\frac{\tau}{4})^{\alpha-1}\mathbb{1}_{\|g_k - \nabla f\| \geq \tau/4}]$ . We further notice that

$$\mathbb{E}[\langle \nabla f, \tau g_k/\|g_k\| \rangle \mathbb{1}_{\{\|g_k\| \geq \tau\}}]$$
$$\geq \tau((1-p)\|\nabla f\|/3 - \frac{8}{3}\mathbb{E}[\|\nabla f - g_k\|])$$
$$\geq \tau(\frac{1-p}{3}\|\nabla f(x)\| - \frac{8\sigma}{3}) \geq \tau(\frac{1-p}{3}\|\nabla f(x)\| - \frac{2\|\nabla f\|}{9}) \tag{3}$$

The last inequality follows by $\|\nabla f(x)\| \geq \tau/2 \geq 12\sigma$. With the above, we get

$$\mathbb{E}[\langle \nabla f, \hat{g}_k \rangle] = \mathbb{E}[\langle \nabla f, g_k \rangle \mathbb{1}\{\|g_k\| \leq \tau\}] + \tau\mathbb{E}[\langle \nabla f, g_k/\|g_k\| \rangle \mathbb{1}\{\|g_k\| \geq \tau\}]$$
$$\geq \frac{p}{2}\|\nabla f\|^2 - \|\nabla f\|\frac{\sigma^\alpha}{(\tau/4)^{\alpha-1}} + \tau(\frac{1-p}{3}\|\nabla f(x)\| - \frac{2\|\nabla f\|}{9})$$
$$\geq \frac{2\tau}{9}\|\nabla f(x)\| - \frac{1}{12}\|\nabla f(x)\| \geq \|\nabla f\|/12$$

The second line follows by Lemma 11 and (2). The third line follows by that $\tau \geq 2$, and $\|\nabla f\| \geq \tau/2$ imply $\frac{p}{2}\|\nabla f\|^2 \geq p\|\nabla f\|/3$. Then, by $\tau \geq 48^{1/(\alpha-1)}\sigma^{\alpha/(\alpha-1)}$, we have $\frac{\sigma^\alpha}{(\tau/4)^{\alpha-1}} \leq \frac{1}{12}$. By $\tau \geq 48\sigma$, $\frac{8}{3}\sigma \leq \tau/12 \leq \|\nabla f\|/6$.

$$\mathbb{E}[f(x_k)] \leq f(x_{k-1}) + \mathbb{E}[\langle \nabla f(x_{k-1}), -\eta_{k-1}\hat{g}_k \rangle] + \frac{\eta_{k-1}^2 L}{2}\tau^2$$
$$\leq f(x_{k-1}) - \eta_{k-1}\|\nabla f(x_{k-1})\|/12 + \eta_{k-1}^2 L\tau\|\nabla f(x_{k-1})\|$$
$$\leq f(x_{k-1}) - \eta_{k-1}\|\nabla f(x_{k-1})\|/24$$

The last inequality above follows by $\eta_k \leq \frac{1}{24L\tau}$.

Combine the two cases we have

$$\mathbb{E}[f(x_k)|x_{k-1}] \leq f(x_{k-1}) - \frac{\eta}{24}\min\{\|\nabla f(x_{k-1})\|^2, \|\nabla f(x_{k-1})\|\} + 2\eta^2 L\sigma^\alpha \tau^{2-\alpha} + \frac{2\eta\sigma^{2\alpha}}{\tau^{2\alpha-2}}.$$

Rearrange and sum the terms above for some fixed step-size and threshold $\{\tau_k = \tau\}$ to get

$$\frac{1}{K}\sum_{k=1}^{K}\mathbb{E}\big[\min\{\|\nabla f(x_{k-1})\|^2, \|\nabla f(x_{k-1})\|\}\big] \leq \frac{24}{\eta K}(f(x_0) - \mathbb{E}[f(x_K)]) + 48\eta L\sigma^\alpha \tau^{2-\alpha} + 48\frac{\sigma^{2\alpha}}{\tau^{2\alpha-2}}$$

$$\leq \underbrace{\frac{24}{\eta K}(f(x_0) - f^\star)}_{T_1} + \underbrace{48\eta L\sigma^\alpha \tau^{2-\alpha} + \frac{48\sigma^{2\alpha}}{\tau^{2\alpha-2}}}_{T_2}.$$

Since we use a stepsize $\eta \leq \frac{1}{L\tau^\alpha}$, we can simplify $T_2$ as

$$\eta L\sigma^\alpha \tau^{2-\alpha} + \frac{\sigma^{2\alpha}}{\tau^{2\alpha-2}} \leq \frac{\sigma^{2\alpha} + \sigma^\alpha}{\tau^{2\alpha-2}}.$$

Denote $F_0 = f(x_0) - f^\star$ to ease notation. Then, adding $T_2$ back to $T_1$ and using a threshold $\tau \geq \sigma K^{\frac{1}{3\alpha-2}}$ we get

$$T_1 + T_2 \leq \frac{24F_0}{K}(L\tau^\alpha + 4L + 24L\tau) + 48\frac{\sigma^{2\alpha} + \sigma^\alpha}{\tau^{2\alpha-2}}$$

$$\leq 48(\sigma^2 + \sigma^{2-\alpha})K^{\frac{-2\alpha+2}{3\alpha-1}}$$

$$+ 24F_0 LK^{-1}(4 + \max\{4, 48^{\alpha/(\alpha-1)}\sigma^{\alpha^2/(\alpha-1)}, 64\sigma^\alpha, \sigma^\alpha K^{\frac{\alpha}{3\alpha-2}}\})$$

$$+ 24F_0 LK^{-1}(\max\{2, 48^{1/(\alpha-1)}\sigma^{\alpha/(\alpha-1)}, 8\sigma, \sigma K^{\frac{1}{3\alpha-2}}\})$$

$$= \mathcal{O}(K^{\frac{-2\alpha+2}{3\alpha-1}})$$

This proves the statement of the theorem. $\qquad\square$

## D  Strongly-Convex Rates (Proof of Theorem 4)

For simplicity, we denote $\hat{g}_k = \min\{\frac{\tau_k}{\|g_k\|}, 1\}g_k$ and the bias $b_k = \mathbb{E}[\hat{g}_k] - \nabla f(x_k)$.

$$\|x_k - x^*\|^2 = \|\operatorname{proj}_{\mathcal{X}}(x_{k-1} - \eta_{k-1}\hat{g}_{k-1} - x^*)\|^2$$

$$\leq \|(x_{k-1} - \eta_{k-1}\hat{g}_{k-1} - x^*)\|^2$$

$$= \|x_{k-1} - x^*\|^2 - 2\eta_{k-1}\langle x_{k-1} - x^*, \nabla f(x_{k-1})\rangle$$

$$- 2\eta_{k-1}\langle x_{k-1} - x^*, b_{k-1}\rangle + \eta_{k-1}^2\|\hat{g}_{k-1}\|^2$$

$$\leq (1 - \mu\eta_{k-1})\|x_{k-1} - x^*\|^2 - 2\eta_{k-1}(f(x_{k-1}) - f^*))$$

$$+ 2\eta_{k-1}(\frac{\mu}{4}\|x_{k-1} - x^*\|^2 + \frac{4}{\mu}\|b_k\|^2) + \eta_{k-1}^2\|\hat{g}_{k-1}\|^2.$$

The first inequality follows by the nonexpansivity of projections onto convex sets.

Rearrange and we get

$$f(x_{k-1}) - f^* \leq \frac{\eta_{k-1}^{-1} - \mu/2}{2}\|x_{k-1} - x^*\|^2 - \frac{\eta_{k-1}^{-1}}{2}\|x_k - x^*\|^2 + \frac{4}{\mu}\|b_k\|^2 + \frac{\eta_{k-1}}{2}\|\hat{g}_{k-1}\|^2.$$

After taking expectation and apply the inequality from Lemma 9, we get

$$\mathbb{E}[f(x_{k-1})] - f^* \leq \mathbb{E}\left[\frac{\eta_{k-1}^{-1} - \mu/2}{2}\|x_{k-1} - x^*\|^2 - \frac{\eta_{k-1}^{-1}}{2}\|x_k - x^*\|^2\right]$$

$$+ 4G^{2\alpha}\tau^{2-2\alpha}\mu^{-1} + \eta_{k-1}G^\alpha\tau^{2-\alpha}/2.$$

Then take $\eta_{k-1} = \frac{4}{\mu(k+1)}, \tau_k = Gk^{\frac{1}{\alpha}}$ and multiply both side by $k$, we get

$$k\mathbb{E}[f(x_{k-1})] - f^* \leq \frac{\mu}{8}\mathbb{E}\big[k(k-1)\|x_{k-1} - x^*\|^2 - k(k+1)\|x_k - x^*\|^2\big]$$

$$+ 8G^2 k^{\frac{2-\alpha}{\alpha}}\mu^{-1}.$$

Notice that $\sum_{k=1}^{K} k^{\frac{2-\alpha}{\alpha}} \leq \int_0^{K+1} k^{\frac{2-\alpha}{\alpha}} dk \leq (K+1)^{2/\alpha}$. Sum over $k$ and we get

$$\sum_{k=1}^{K} k\mathbb{E}[f(x_{k-1})] - f^* \leq \frac{\mu}{8}\mathbb{E}\big[-T(T+1)\|x_T - x^*\|^2\big]$$
$$+ 8G^2(K+1)^{\frac{2}{\alpha}}\mu^{-1}.$$

Devide both side by $\frac{K(K+1)}{2}$ and we get

$$\frac{2}{K(K+1)} \sum_{k=1}^{K} k\mathbb{E}[f(x_{k-1})] - f^* \leq 8G^2 K^{-1}(K+1)^{\frac{2-\alpha}{\alpha}}\mu^{-1}.$$

Notice that for $K \geq 1$, $K^{-1} \leq 2(K+1)^{-1}$. We have

$$\frac{2}{K(K+1)} \sum_{k=1}^{K} k\mathbb{E}[f(x_{k-1})] - f^* \leq 16G^2(K+1)^{\frac{2-2\alpha}{\alpha}}\mu^{-1}.$$

The theorem then follows by Jensen's inequality. $\qquad\square$

# E    Effect of coordinate-wise moment bound

We now examine how the rates would change if we replace Assumption 4 with Assumption 2.

## E.1    Convergence of GClip (proof of Corollary 7)

We now look at(GClip) under assumption 2.

The proof of both the convex and non-convex rates following directly from the following Lemma.

**Lemma 12.** *For any $g(x)$ suppose that assumption 2 with $\alpha \in (1,2]$. Then suppose we have a constant upper-bound*

$$\mathbb{E}[\|g(x)\|^\alpha] \leq D.$$

*Then $D$ satisfies*

$$d^{\frac{\alpha}{2}-1}\|B\|_\alpha^\alpha \leq D \leq d^{\alpha/2}\|B\|_\alpha^\alpha.$$

*Proof.* Note that the function $(\cdot)^{\alpha/2}$ is concave for $\alpha \in (1,2]$. Using Jensen's inequality we can rewrite as:

$$D \geq \mathbb{E}[\|g(x)\|^\alpha] = d^{\alpha/2}\mathbb{E}\left[\left(\frac{1}{d}\sum_{i=1}^{d}|g(x)^{(i)}|^2\right)^{\alpha/2}\right] \geq d^{\alpha/2-1}\mathbb{E}\left[\sum_{i=1}^{d}|g(x)^{(i)}|^\alpha\right].$$

Since the right hand-side can be as large as $d^{\frac{\alpha}{2}-1}\|B\|_\alpha^\alpha$, we have our first inequality. On the other hand, we also have an upper bound below:

$$\mathbb{E}[\|g(x)\|^\alpha] = \mathbb{E}\left[\left(\sum_{i=1}^{d}|g(x)^{(i)}|^2\right)^{\alpha/2}\right] \leq \mathbb{E}\left[\left(d(\max_{i=1}^{d} g(x)^{(i)})^2\right)^{\alpha/2}\right]$$

$$\leq \mathbb{E}\left[d^{\alpha/2}(\max_{i=1}^{d} g(x)^{(i)})^\alpha\right] \leq \mathbb{E}\left[d^{\alpha/2}\sum_{i=1}^{d}(g(x)^{(i)})^\alpha\right] \leq d^{\alpha/2}\sum_{i=1}^{d}B_i^\alpha$$

where $\|B\|_\alpha^\alpha = \sum_{i=1}^{d} B_i^\alpha$. Thus, we have shown that

$$d^{\frac{\alpha}{2}-1}\|B\|_\alpha^\alpha \leq \mathbb{E}[\|g(x)\|^\alpha] \leq d^{\alpha/2}\|B\|_\alpha^\alpha.$$

We know that Jensen's inequality is tight when all the co-ordinates have equal values. This means that if the noise across the coordinates is linearly correlated the lower bound is tighter, whereas the upper bound is tighter if the coordinates depend upon each other in a more complicated manner or are independent of each other. $\qquad\square$

Substituting this bound on $G$ in Theorems 4 and 2 gives us our corollaries.

## E.2 Convergence of CClip (Proof of Theorem 8)

The proof relies on the key lemma which captures the bias-variance trade off under the new noise-assumption and coordinate-wise clipping.

**Lemma 13.** *For any $g(x)$ suppose that assumption 2 with $\alpha \in (1,2]$ holds. Denote $g_i$ to be $i_{th}$ component of $g(x)$, $\nabla f(x)_i$ to be $i_{th}$ component of $\nabla f(x)$. Then the estimator $\hat{g}(x) = [\hat{g}_1; \cdots ; \hat{g}_d]$ from (CClip) with clipping parameter $\tau = [\tau_1; \tau_2; \cdots ; \tau_d]$ satisfies:*

$$\mathbb{E}\big[\|\hat{g}_i\|^2\big] \le B_i^\alpha \tau_i^{2-\alpha} \text{ and } \|\mathbb{E}[\hat{g}_i] - \nabla f(x)_i\|^2 \le B_i^{2\alpha} \tau_i^{-2(\alpha-1)}.$$

*Proof.* Apply Lemma 9 to the one dimensional case in each coordinate. □

*Proof of Theorem 8.* Theorem 4 For simplicity, we denote $\hat{g}_k = \eta_{k-1}\hat{g}(x_k)$ and the bias $b_k = \mathbb{E}[\hat{g}_k] - \nabla f(x_k)$.

$$\begin{aligned}
\|x_k - x^*\|^2 &= \|x_{k-1} - \eta_{k-1}\hat{g}_{k-1} - x^*\|^2 \\
&= \|x_{k-1} - x^*\|^2 - 2\eta_{k-1}\langle x_{k-1} - x^*, \nabla f(x_{k-1})\rangle \\
&\quad - 2\eta_k\langle x_{k-1} - x^*, b_{k-1}\rangle + \eta_k^2\|\hat{g}_{k-1}\|^2 \\
&\le (1 - \mu\eta_k)\|x_{k-1} - x^*\|^2 - 2\eta_k(f(x_{k-1}) - f^*)) \\
&\quad + 2\eta_k(\frac{\mu}{4}\|x_{k-1} - x^*\|^2 + \frac{4}{\mu}\|b_k\|^2) + \eta_k^2\|\hat{g}_{k-1}\|^2.
\end{aligned}$$

Rearrange and we get

$$f(x_{k-1}) - f^* \le \frac{\eta_k^{-1} - \mu/2}{2}\|x_{k-1} - x^*\|^2 - \frac{\eta_k^{-1}}{2}\|x_k - x^*\|^2 + \frac{4}{\mu}\|b_k\|^2 + \frac{\eta_k}{2}\|\hat{g}_{k-1}\|^2.$$

After taking expectation and apply the inequality from Lemma 9, we get

$$\begin{aligned}
\mathbb{E}[f(x_{k-1})] - f^* &\le \mathbb{E}\left[\frac{\eta_k^{-1} - \mu/2}{2}\|x_{k-1} - x^*\|^2 - \frac{\eta_k^{-1}}{2}\|x_k - x^*\|^2\right] \\
&\quad + \sum_{i=1}^d 4B_i^{2\alpha}\tau_i^{2-2\alpha}\mu^{-1} + \eta_k G^\alpha \tau_i^{2-\alpha}/2.
\end{aligned}$$

Then take $\eta_k = \frac{4}{\mu(k+1)}, \tau_i = B_i k^{\frac{1}{\alpha}}$ and multiply both side by $k$, we get

$$\begin{aligned}
k\mathbb{E}[f(x_{k-1})] - f^* &\le \frac{\mu}{8}\mathbb{E}\big[k(k-1)\|x_{k-1} - x^*\|^2 - k(k+1)\|x_k - x^*\|^2\big] \\
&\quad + 8\sum_{i=1}^d B_i^2 k^{\frac{2-\alpha}{\alpha}}\mu^{-1}.
\end{aligned}$$

Notice that $\sum_{k=1}^K k^{\frac{2-\alpha}{\alpha}} \le \int_0^{K+1} k^{\frac{2-\alpha}{\alpha}} dk \le (K+1)^{2/\alpha}$. Sum over $k$ and we get

$$\begin{aligned}
\sum_{k=1}^K k\mathbb{E}[f(x_{k-1})] - f^* &\le \frac{\mu}{8}\mathbb{E}\big[-T(T+1)\|x_T - x^*\|^2\big] \\
&\quad + 8\sum_{i=1}^d B_i^2 k^{\frac{2-\alpha}{\alpha}}\mu^{-1}.
\end{aligned}$$

Devide both side by $\frac{K(K+1)}{2}$ and we get

$$\frac{2}{K(K+1)}\sum_{k=1}^K k\mathbb{E}[f(x_{k-1})] - f^* \le 8\sum_{i=1}^d B_i^2 k^{\frac{2-\alpha}{\alpha}}\mu^{-1}.$$

Notice that for $K \ge 1$, $K^{-1} \le 2(K+1)^{-1}$. We have

$$\frac{2}{K(K+1)}\sum_{k=1}^K k\mathbb{E}[f(x_{k-1})] - f^* \le 16\sum_{i=1}^d B_i^2 k^{\frac{2-\alpha}{\alpha}}\mu^{-1}.$$

The theorem then follows by Jensen's inequality. □ □

# F    Lower Bound (Proof of Theorem 5)

We consider the following simple one-dimensional function class parameterized by $b$:

$$\min_{x \in [0,1/2]} \left\{ f_b(x) = \tfrac{1}{2}(x - b)^2 \right\}, \text{ for } b \in [0, 1/2]. \tag{4}$$

Also suppose that for $\alpha \in (1, 2]$ and $b \in [0, 1/2]$ the stochastic gradients are of the form:

$$g(x) \sim \nabla f_b(x) + \chi_b, \, \mathbb{E}[g(x)] = \nabla f_b(x), \text{ and } \mathbb{E}[|g(x)|^\alpha] \le 1. \tag{5}$$

Note that the function class (4) has $\mu = 1$ and optimum value $f_b(b) = 0$, and the $\alpha$-moment of the noise in (5) satisfies $G = B \le 1$. Thus, we want to prove the following:

**Theorem 14.** *For any $\alpha \in (1, 2]$ there exists a distribution $\chi_b$ such that the stochastic gradients satisfy* (5). *Further, for any (possibly randomized) algorithm $\mathcal{A}$, define $\mathcal{A}_k(f_b + \chi_b)$ to be the output of the algorithm $\mathcal{A}$ after $k$ queries to the stochastic gradient $g(x)$. Then:*

$$\max_{b \in [0,1/2]} \mathbb{E}[f_b(\mathcal{A}_k(f_b + \chi_b))] \ge \Omega\left( \frac{1}{k^{2(\alpha-1)/\alpha}} \right).$$

Our lower bound construction is inspired by Theorem 2 of [4]. Let $\mathcal{A}_k(f_b + \chi_b)$ denote the output of any possibly randomized algorithm $\mathcal{A}$ after processing $k$ stochastic gradients of the function $f_b$ (with noise drawn i.i.d. from distribution $\chi_b$). Similarly, let $\mathcal{D}_k(f_b + \chi_b)$ denote the output of a *deterministic* algorithm after processing the $k$ stochastic gradients. Then from Yao's minimax principle we know that for any fixed distribution $\mathcal{B}$ over $[0, 1/2]$,

$$\min_{\mathcal{A}} \max_{b \in [0,1/2]} \mathbb{E}_{\mathcal{A}}[\mathbb{E}_{\chi_b} f_b(\mathcal{A}_k(f_b + \chi_b))] \ge \min_{\mathcal{D}} \mathbb{E}_{b \sim \mathcal{B}}[\mathbb{E}_{\chi_b} f_b(\mathcal{D}_k(f_b + \chi_b))].$$

Here we denote $\mathbb{E}_{\mathcal{A}}$ to be expectation over the randomness of the algorithm $\mathcal{A}$ and $\mathbb{E}_{\chi_b}$ to be over the stochasticity of the the noise distribution $\chi_b$. Hence, we only have to analyze deterministic algorithms to establish the lower-bound. Further, since $\mathcal{D}_k$ is deterministic, for any *bijective* transformation $h$ which transforms the stochastic gradients, there exists a deterministic algorithm $\tilde{\mathcal{D}}$ such that $\tilde{\mathcal{D}}_k(h(f_b + \chi_b)) = \mathcal{D}_k(f_b + \chi_b)$. This implies that for any bijective transformation $h(\cdot)$ of the gradients:

$$\min_{\mathcal{D}} \mathbb{E}_{b \sim \mathcal{B}}[\mathbb{E}_{\chi_b} f_b(\mathcal{D}_k(f_b + \chi_b))] = \min_{\mathcal{D}} \mathbb{E}_{b \sim \mathcal{B}}[\mathbb{E}_{\chi_b} f_b(\mathcal{D}_k(h(f_b + \chi_b)))].$$

In this rest of the proof, we will try obtain a lower bound for the right hand side above.

We now describe our construction of the three quantities to be defined: the problem distribution $\mathcal{B}$, the noise distribution $\chi_b$, and the bijective mapping $h(\cdot)$. All of our definitions are parameterized by $\alpha \in (1, 2]$ (which is given as input) and by $\epsilon \in (0, 1/8]$ (which represents the desired target accuracy). We will pick $\epsilon$ to be a fixed constant which depends on the problem parameters (e.g. $k$) and should be thought of as being small.

- Problem distribution: $\mathcal{B}$ picks $b_0 = 2\epsilon$ or $b_1 = \epsilon$ at random i.e. $\nu \in \{0, 1\}$ is chosen by an unbiased coin toss and then we pick

$$b_\nu = (2 - \nu)\epsilon. \tag{6}$$

- Noise distribution: Define a constant $\gamma = (4\epsilon)^{1/(\alpha-1)}$ and $p_\nu = (\gamma^\alpha - 2\nu\gamma\epsilon)$. Simple computations verify that $\gamma \in (0, 1/2]$ and that

$$p_\nu = (4\epsilon)^{\frac{\alpha}{\alpha-1}} - 2\nu(4\epsilon^\alpha)^{\frac{1}{\alpha-1}} = (4 - 2\nu)(4\epsilon^\alpha)^{\frac{1}{\alpha-1}} \in (0, 1).$$

Then, for a given $\nu \in \{0, 1\}$ the stochastic gradient $g(x)$ is defined as

$$g(x) = \begin{cases} x - \frac{1}{2\gamma} & \text{with prob. } p_\nu, \\ x & \text{with prob. } 1 - p_\nu. \end{cases} \tag{7}$$

To see that we have the correct gradient in expectation verify that

$$\mathbb{E}[g(x)] = x - \frac{p_\nu}{2\gamma} = x - \frac{\gamma^{\alpha-1}}{2} + \nu\epsilon = x - (2 - \nu)\epsilon = x - b_\nu = \nabla f_{b_\nu}(x).$$

Next to bound the $\alpha$ moment of $g(x)$ we see that

$$\mathbb{E}[|g(x)|^\alpha] \le \gamma^\alpha \left( x - \frac{1}{2\gamma} \right)^\alpha + x^\alpha \le \frac{1}{2} + \frac{1}{2} = 1 \,.$$

The above inequality used the bounds that $\alpha \ge 1$, $x \in [0, 1/2]$, and $\gamma \in (0, 1/2]$. Thus $g(x)$ defined in (7) satisfies condition (5).

- Bijective mapping: Note that here the only unknown variable is $\nu$ which only affects $p_\nu$. Thus the mapping is bijective as long as the *frequencies* of the events are preserved. Hence given a stochastic gradient $g(x_i)$ the mapping we use is:

$$h(g(x_i)) = \begin{cases} 1 & \text{if } g(x_i) = x_i - \frac{1}{2\gamma} \,, \\ 0 & \text{otherwise.} \end{cases} \tag{8}$$

Given the definitions above, the output of algorithm $\mathcal{D}_k$ is thus simply a function of $k$ i.i.d. samples drawn from the Bernoulli distribution with parameter $p_\nu$ (which is denoted by $\text{Bern}(p_\nu)$). We now show how achieving a small optimization error implies being able to guess the value of $\nu$.

**Lemma 15.** *Suppose we are given problem and noise distributions defined as in (6) and (7), and an bijective mapping $h(\cdot)$ as in (8). Further suppose that there is a deterministic algorithm $\mathcal{D}_k$ whose output after processing $k$ stochastic gradients satisfies*

$$\mathbb{E}_{b \sim \mathcal{B}}[\mathbb{E}_{\chi_b} f_b(\mathcal{D}_k(h(f_b + \chi_b)))] < \epsilon^2/64 \,.$$

*Then, there exists a deterministic function $\tilde{\mathcal{D}}_k$ which given $k$ independent samples of $\text{Bern}(p_\nu)$ outputs $\nu' = \tilde{\mathcal{D}}_k(\text{Bern}(p_\nu)) \in \{0, 1\}$ such that*

$$\Pr\left[ \tilde{\mathcal{D}}_k(\text{Bern}(p_\nu)) = \nu \right] \ge \frac{3}{4} \,.$$

*Proof.* Suppose that we are given access to $k$ samples of $\text{Bern}(p_\nu)$. Use these $k$ samples as the input $h(f_b + \chi_b))$ to the procedure $\mathcal{D}_k$ (this is valid as previously discussed), and let the output of $\mathcal{D}_k$ be $x_k^{(\nu)}$. The assumption in the lemma states that

$$\mathbb{E}_\nu \left[ \mathbb{E}_{\chi_b} |x_k^{(\nu)} - b_\nu|^2 \right] < \frac{\epsilon^2}{32}, \text{ which implies that } \mathbb{E}_{\chi_b} |x_k^{(\nu)} - b_\nu|^2 < \frac{\epsilon^2}{16} \text{ almost surely.}$$

Then, using Markov's inequality (and then taking square-roots on both sides) gives

$$\Pr\left[ |x_k^{(\nu)} - b_\nu| \ge \frac{\epsilon}{2} \right] \le \frac{1}{4} \,.$$

Consider a simple procedure $\tilde{\mathcal{D}}_k$ which outputs $\nu' = 0$ if $x_k^{(\nu)} \ge \frac{3\epsilon}{2}$, and $\nu' = 1$ otherwise. Recall that $|b_0 - b_1| = \epsilon$ with $b_0 = 2\epsilon$ and $b_1 = \epsilon$. With probability $\frac{3}{4}$, $|x_k^{(\nu)} - b_\nu| < \frac{\epsilon}{2}$ and hence the output $\nu'$ is correct. $\square$

Lemma 15 shows that if the optimization error of $\mathcal{D}_k$ is small, there exists a procedure $\tilde{\mathcal{D}}_k$ which distinguishes between the Bernoulli distributions with parameters $p_0$ and $p_1$ using $k$ samples. To argue that the optimization error is large, one simply has to argue that a large number of samples are required to distinguish between $\text{Bern}(p_0)$ and $\text{Bern}(p_1)$.

**Lemma 16.** *For any deterministic procedure $\tilde{\mathcal{D}}_k(\text{Bern}(p_\nu))$ which processes $k$ samples of $\text{Bern}(p_\nu)$ and outputs $\nu'$*

$$\Pr[\nu' = \nu] \le \frac{1}{2} + \sqrt{k(4\epsilon)^{\frac{\alpha}{\alpha-1}}} \,.$$

*Proof.* Here it would be convenient to make the dependence on the samples explicit. Denote $_k^{(\nu)} = \left( s_1^{(\nu)}, \dots, s_k^{(\nu)} \right) \in \{0, 1\}^k$ to be the $k$ samples drawn from $\text{Bern}(p_\nu)$ and denote the output as $\nu' = \tilde{\mathcal{D}}(_k^{(\nu)})$. With some slight abuse of notation where we use the same symbols to denote the realization and their distributions, we have:

$$\Pr\left[ \tilde{\mathcal{D}}(_k^{(\nu)}) = \nu \right] = \frac{1}{2} \Pr\left[ \tilde{\mathcal{D}}(_k^{(1)}) = 1 \right] + \frac{1}{2} \Pr\left[ \tilde{\mathcal{D}}(_k^{(0)}) = 0 \right] = \frac{1}{2} + \frac{1}{2} \mathbb{E}\left[ \tilde{\mathcal{D}}(_k^{(1)}) - \tilde{\mathcal{D}}(_k^{(0)}) \right] \,.$$

Next using Pinsker's inequality we can upper bound the right hand side as:

$$\mathbb{E}\left[\tilde{\mathcal{D}}\binom{(1)}{k} - \tilde{\mathcal{D}}\binom{(0)}{k}\right] \leq \left|\tilde{\mathcal{D}}\binom{(1)}{k} - \tilde{\mathcal{D}}\binom{(0)}{k}\right|_{TV} \leq \sqrt{\frac{1}{2}\operatorname{KL}\left(\tilde{\mathcal{D}}\binom{(1)}{k}, \tilde{\mathcal{D}}\binom{(0)}{k}\right)},$$

where $|\cdot|_{TV}$ denotes the total-variation distance and $\operatorname{KL}(\cdot, \cdot)$ denotes the KL-divergence. Recall two properties of KL-divergence: i) for a product measures defined over the same measurable space $(p_1, \ldots, p_k)$ and $(q_1, \ldots, q_k)$,

$$\operatorname{KL}((p_1, \ldots, p_k), (q_1, \ldots, q_k)) = \sum_{i=1}^{k} \operatorname{KL}(p_i, q_i),$$

and ii) for any deterministic function $\tilde{\mathcal{D}}$,

$$\operatorname{KL}(p, q) \geq \operatorname{KL}(\tilde{\mathcal{D}}(p), \tilde{\mathcal{D}}(q)).$$

Thus, we can simplify as

$$\begin{aligned}
\Pr\left[\tilde{\mathcal{D}}\binom{(\nu)}{k} = \nu\right] &\leq \frac{1}{2} + \sqrt{\frac{k}{8}\operatorname{KL}(\operatorname{Bern}(p_1), \operatorname{Bern}(p_0))} \\
&\leq \frac{1}{2} + \sqrt{\frac{k}{8}\frac{(p_0 - p_1)^2}{p_0(1 - p_0)}} \\
&\leq \frac{1}{2} + \sqrt{\frac{k(\gamma\epsilon)^2}{4\gamma^\alpha}} \\
&= \frac{1}{2} + \sqrt{k\left(4^{(2-1/\alpha)}\epsilon\right)^{\frac{\alpha}{\alpha-1}}}.
\end{aligned}$$

Recalling that $\alpha \in (1, 2]$ gives us the statement of the lemma. $\qquad\square$

If we pick $\epsilon$ to be

$$\epsilon = \frac{1}{16k^{(\alpha-1)/\alpha}},$$

we have that

$$\frac{1}{2} + \sqrt{k(4\epsilon)^{\frac{\alpha}{\alpha-1}}} < \frac{3}{4}.$$

Given Lemmas 15 and 16, this implies that for the above choice of $\epsilon$,

$$\mathbb{E}_{b \sim \mathcal{B}}[\mathbb{E}_{\chi_b} f_b(\mathcal{D}_k(h(f_b + \chi_b)))] \geq \epsilon^2/64 = \frac{1}{2^{14}k^{2(\alpha-1)/\alpha}}.$$

This finishes the proof of the theorem. Note that the readability of the proof was prioritized over optimality and it is possible to obtain significantly better constants. $\qquad\square$

# G    Non-convex Lower Bound (Proof of Theorem 6)

The proof is based on the proof of Theorem 1 in [2]. The only difference is that we assume bounded $\alpha-$moment of the stochastic oracle instead of bounded variance as in the original proof. We refer readers to [2] for more backgrounds and intuitions. For convenience, we study the stochastic setting ($K = 1$ in [2]) instead of batched setting. We denote a $d-$dimensional vector $x$ as, $x = [x_{(1)}; \ldots; x_{(d)}]$. Let support$(x)$ denote the set of coordinates where $x$ is nonzero, i.e.

$$\operatorname{support}(x) = \{i \in [d] | x_{(i)} \neq 0\} \subseteq [d].$$

Denote $\operatorname{prog}_\beta(x)$ as the highest index whose entry is $\beta-$far from zero.

$$\operatorname{prog}_\beta(x) = \max\{i \in [d] | |x_{(i)}| > \beta\} \in [d].$$

Note that the function $\mathrm{prog}_\beta(\,\cdot\,)$ is decreasing in $\beta$. The function we use to prove the theorem is the same as in [2, 5]. We denote

$$f_d(x) = -\Psi(1)\Phi(x_{(1)}) + \sum_{i=2}^{d}\big(\Psi(-x_{(i-1)})\Phi(-x_{(i)}) - \Psi(x_{(i-1)})\Phi(x_{(i)})\big), \text{ where}$$

$$\Psi(x) = \begin{cases} 0, & x \le 1/2 \\ \exp(1 - \frac{1}{(2x-1)^2}), x > 1/2 \end{cases}, \Phi(x) = \sqrt{e}\int_{-\infty}^{x} e^{-\frac{t^2}{2}}\,dt.$$

The above function satisfies the following important properties,

**Lemma 17** (Lemma 2 in [2]). *The function $f_d$ satisfies the following properties,*

1. $f_d(0) - \inf_x f_d(x) \le 12d$.

2. $f_d$ is $L_0$-smooth, where $L_0 = 152$.

3. For all $x$, $\|\nabla f_d(x)\|_\infty \le 23$.

4. For all $x$, $\mathrm{prog}_0(\nabla f_d(x)) \le \mathrm{prog}_{\frac12}(x) + 1$

5. For all $x$, if $\mathrm{prog}_1(x) < d$, then $\|\nabla f_d(x)\|^2 \ge 1$.

We also define the stochastic oracle $g_d(x)$ as below

$$g_d(x)_{(i)} = \left(1 + \mathbb{1}\Big\{i = \mathrm{prog}_{\frac14}(x) + 1\Big\}\Big(\frac{z}{p} - 1\Big)\right)\frac{\partial}{\partial x_{(i)}}f_d(x)$$

where $z \sim \text{Bernoulli}(p)$. The stochasticity of $g_d(x)$ is only in the $(\mathrm{prog}_{\frac14}(x) + 1)$th coordinate. It is easy to see that $g_d(x)$ is a probability-$p$ zero chain as in [2, Definition 2] i.e. it satisfies

$$\mathbb{P}\Big(\exists x, \text{ s.t. } \mathrm{prog}_0(g_d(x)) = \mathrm{prog}_{\frac14}(x) + 1\Big) \le p,$$

$$\mathbb{P}\Big(\exists x, \text{ s.t. } \mathrm{prog}_0(g_d(x)) > \mathrm{prog}_{\frac14}(x) + 1\Big) = 0.$$

The second claim is because $\mathrm{prog}_\beta(\,\cdot\,)$ is decreasing in $\beta$ and

$$\mathrm{prog}_{\frac14}(\nabla f_d(x)) \le \mathrm{prog}_0(\nabla f_d(x)) \le \mathrm{prog}_{\frac12}(x) + 1 \le \mathrm{prog}_{\frac14}(x) + 1\,.$$

The first claim is because if $z = 0$, then we explicitly set the $(\mathrm{prog}_{\frac14}(x) + 1)$th coordinate to 0. The stochastic gradient additionally has bounded $\alpha$-moment as we next show.

**Lemma 18.** *The stochastic oracle above is an unbiased estimator of the true gradient, and for any $\alpha \in (1, 2]$*

$$\mathbb{E}[\|g_d(x)\|^\alpha] \le 2\|\nabla f_d(x)\|^\alpha + 23^\alpha\frac{2}{p^{\alpha-1}}.$$

*Proof.* The unbiased-ness is easy to verify. For the bounded $\alpha$-moment, observe that only the $(\mathrm{prog}_{\frac14} + 1)$-th coordinate is noisy and differs by a factor of $(\frac{z}{p} - 1)$. Hence, we have

$$\mathbb{E}[\|g_d(x)\|^\alpha] \le 2\|\nabla f_d(x)\|^\alpha + 2\mathbb{E}[\|g_d(x) - \nabla f_d(x)\|^\alpha]$$

$$\le 2\|\nabla f_d(x)\|^\alpha + \|\nabla f_d(x)\|_\infty^\alpha\mathbb{E}\Big[|\frac{z}{p} - 1|^\alpha\Big]$$

$$\le 2\|\nabla f_d(x)\|^\alpha + \|\nabla f_d(x)\|_\infty^\alpha\frac{p(1-p)^\alpha + (1-p)p^\alpha}{p^\alpha}$$

$$\le 2\|\nabla f_d(x)\|^\alpha + 23^\alpha\frac{2}{p^{\alpha-1}}$$

The first inequality followed from Jensen's inequality and the convexity of $\|\cdot\|^\alpha$ for $\alpha \in (1, 2]$:

$$\|u + v\|^\alpha \le 4\|\tfrac{u+v}{2}\|^\alpha \le 2(\|u\|^\alpha + \|v\|^\alpha) \text{ for any } u, v\,.$$

$\square$

Now we are ready to prove Theorem 6. Given accuracy parameter $\epsilon$, suboptimality $\Delta = f(0) - f^*$, smoothness constant $L$, and bounded $\alpha-$moment $G^\alpha$, we define

$$f(x) = \frac{L\lambda^2}{152}f_d\left(\frac{x}{\lambda}\right),$$

where $\lambda = \frac{304\epsilon}{L}$ and $d = \lfloor \frac{\Delta L}{7296\epsilon^2}\rfloor$. Then,

$$g(x) = \frac{L\lambda}{152}g_d(x/\lambda) = 2\epsilon g_d(x/\lambda).$$

Using Lemma 18, we have

$$\mathbb{E}[\|g(x)\|^\alpha] \leq 8\epsilon^\alpha\|\nabla f_d(x)\|^\alpha + \frac{5000\epsilon^\alpha}{p^{\alpha-1}}$$

When $G \geq 4\sqrt{\Delta L}$, we can set $p = \frac{(5000\epsilon)^{\frac{\alpha}{\alpha-1}}}{(G-4\sqrt{\Delta L})^{\frac{\alpha}{\alpha-1}}}$ and get $\mathbb{E}[\|g(x)\|^\alpha] \leq G^\alpha$.

Let $x_k$ be the output of any *zero-respecting* algorithm $\mathcal{A}$. By [2, Lemma 1], we know that with probability at least 1/2, $\text{prog}_1(x_k) \leq \text{prog}_0(x_k) < d$ for all $k \leq \frac{(d-1)}{2p}$. Now applying Lemma 17.5, we have that for all $k \leq \frac{(d-1)}{2p}$:

$$\mathbb{E}[\|\nabla f(x_k)\|] \geq \frac{1}{2}\frac{L\lambda}{152}\mathbb{E}[\mathbb{1}\|\nabla f_d(x_k/\lambda)\| \,|\, \{\text{prog}_1(x_k) < d\}] \geq \epsilon\,.$$

Therefore, $\mathbb{E}\|\nabla f(x_k)\| \geq \epsilon$, for all $k \leq \frac{(d-1)}{2p} = \frac{(G-4\sqrt{\Delta L})^{\frac{\alpha}{\alpha-1}}\Delta L}{7296\times 5000^{\frac{\alpha}{\alpha-1}}\epsilon^{2+\frac{\alpha}{\alpha-1}}} = c(\alpha)(G - 4\sqrt{\Delta L})^{\frac{\alpha}{\alpha-1}}\Delta L\epsilon^{-\frac{3\alpha-2}{\alpha-1}}$. By eliminating $\epsilon$, we can rewrite this in terms of $k$. Finally, the techniques from [2, Theorem 3] show how to lift lower-bounds for *zero-respecting* algorithms to any randomized method.

## H  A Comparison with [25]

We are not the first to study the heavy-tailed noise behavior in neural network training. The novel work by Simsekli et al. [25] studies the noise behavior of AlexNet on Cifar 10 and observed that the noise does not seem to come from Gaussian distribution. However, in our AlexNet training with ImageNet data, we observe that the noise histogram looks Gaussian as in Figure 5(a, b). We believe the difference results from that in [25], the authors treat the noise in each coordinate as an independent scaler noise, as described in the original work on applying tail index estimator. We on the other hand, consider each the noise as a high dimensional random vector computed from a minibatch. We are also able to observe heavy tailed noise if we fix a single minibatch and plot the noise in each dimension, as shown in Figure 5(c). The fact that noise is well concentrated on Cifar is also observed by Panigrahi et al. [20].

Figure 5: (a) Noise histogram of AlexNet on ImageNet data at initialization. (b)Noise histogram of AlexNet on ImageNet data at 5k iterations. (c) The per dimension noise distribution within a single minibatch at initialization.

Furthermore, we used the tail index estimator presented in [25] to estimate the tail index of noise norm distribution. Though some assumptions of the estimator are not satisfied (in our case, the symmetry assumption; in [25], the symmetry assumption and independence assumption), we think it can be an indicator for measuring the "heaviness" of the tail distribution.

(a) ImageNet training, $\hat{\alpha} = 1.99$       (b) Bert pretraining, $\hat{\alpha} = 1.08$

Figure 6: Tail index estimation of gradient noise in ImageNet training and BERT training.

# I   ACClip in ImageNet Training

For completeness, we test ACClip on ImageNet training with ResNet50. After hyperparameter tuning for all algorithms, ACClip is able to achieve better performance compared to ADAM, but worse performance compared to SGD. This is as expected because the noise distribution in ImageNet + ResNet50 training is well concentrated. The validation accuracy for SGD, ADAM, ACClip are $0.754, 0.716, 0.730$ respectively.

(a)

Figure 7: Validation loss for ResNet50 trained on ImageNet. SGD outperforms Adam and ACClip.