[Reviews · NeurIPS 2020]

Review 1

Summary and Contributions: The paper studies the behavior of SGD, Adam, and SGD with clipping on the stochastic optimization problems with heavy-tailed stochastic gradients. First of all, the authors empirically establish that Adam outperforms SGD on the problems with heavy-tailed stochastic gradients. Next, they derive the convergence guarantees for clipped SGD for smooth non-convex under the assumption of the uniformly bounded central moment of order \alpha \in (1,2] of the gradient and non-smooth (authors claim that f should be L-smooth in the statement of the theorem, but do not use it in the proof) strongly convex problems under the assumption of the uniformly bounded moment of order \alpha \in (1,2] of the gradient. Interestingly, in these cases, SGD can diverge, which fits the empirical evidence that methods with clipping (or its adaptive variants) work better than SGD in the presence of heavy-tailed noise. Furthermore, the paper proposes lower bounds for these cases implying the optimality of clipped SGD. Under the coordinate-wise boundedness of \alpha-th moment of the stochastic gradient, authors extended their analysis of clipped SGD (GClip) for the strongly convex case. However, the obtained bound depends linearly on the dimension of the problem. To remove this drawback, authors consider coordinate-wise clipping for SGD (CClip) and derive the dimensional-independent convergence guarantee. Finally, the authors propose an adaptive version of CClip (ACClip) and empirically show its superior over Adam for training attention models. ==========After Reading the Response============= I have read the response and other reviews. First of all, I want to thank the authors for their thorough response addressing my main concerns. I have checked these modifications - they are correct and make the proofs of Remark 1 and Theorem 2 technically correct. However, the second comment in the response (after the sentence "Strongly convex and bounded second moment implies bounded domain") about Theorem 4 does not address the issue properly. I think the modification of the proof showing that it is enough to assume boundedness of the gradient only on some compact/ball is not too straightforward and requires more refined analysis than one presented in the paper. Despite the fact that this is not the main result of the paper, it should be formulated and prooved properly (under reasonable assumptions). The current version is misleading: the authors write in the main part that they are solving the minimization problem on R^d, but the analysis does not work on R^d, and in the appendix, it is stated, "However, the domain diameter is not used explicitly in the proof. The projected clipped gradient descent follows exactly the same argument and the fact that orthogonal projections contract distances," which is about optimization problem on some bounded set. This place requires more clarifications. For example, it is not discussed how this set should be chosen, but it is an important question. Next, R2 and R3 noticed a lot of issues with the experimental part of the paper that I did not write about in my review. For this submission experimental part plays a significant role, so I think the authors should add missing details and explain some parts (like Figure 1) thoroughly. To conclude, I find the paper very interesting, but taking into account the large number of technical issues and inaccuracies noticed by other reviewers and me, the paper should be significantly re-written and, as a consequence, completely re-reviewed. Therefore, I keep my initial score (5) unchanged, and encourage authors to revise the paper, apply needed corrections according to the reviews, and re-submit to another conference or journal.

Strengths: 1. Extensive numerical study of the relationship between good behavior of clipped methods and the presence of the heavy-tailed noise in the stochastic gradients. 2. For the considered setup lower bounds for the convergence rates were obtained.

Weaknesses: 1. The proofs contain a lot of unexplained and inaccurate places. It seems that the paper was written in a rush (see the detailed comments below). Furthermore, one place in the proof of Theorem 2 seems to be incorrect or at least requires a more thorough explanation/proof: the first equality in line 405 is not explained enough. For me, it is not clear how it was obtained. Moreover, it seems to be incorrect. 2. The analysis in the strongly convex case is conducted under the assumption of uniform boundedness of the \alpha-th moment of the stochastic gradient, which implies the boundedness of the gradients of the objective function. This combination of two assumptions cannot hold on the entire space while the analysis works on the whole space. Therefore, a more refined analysis is needed. Moreover, I believe that it should be explicitly discussed in the main text of the paper.

Correctness: It is hard to go through the proofs of some results (in particular, through the second part of Theorem 2) because of many inaccuracies and unexplained claims. As it was mentioned before, the proof of Theorem 2 contains an unexplained place that seems to be incorrect. Proofs of Theorems 4 and 8 seem to be correct and use the classical scheme of the proof for SGD.

Clarity: The paper contains a lot of typos and is hard to follow in some parts (especially in the appendix).

Relation to Prior Work: The relation to the prior work is well explained.

Reproducibility: No

Additional Feedback: As I mentioned before, I was not sure in the correctness of the proof of Theorem 2 which is one of the main contributions of the paper. I encourage the authors to give a detailed response in the rebuttal how the mentioned equality (the first one in the line 405) was obtained and if it is not correct how it could be fixed. Line 99: “we propose” – this algorithm was known before, so, it is not correct that you propose GClip. Line 105: consider renaming as “uniformly bounded \alpha-th central moment” line 116: a typo in the subscript of \|\nabla f(x_{k}+1)\| → \|\nabla f(x_{k+1})\| lines 116-117: I do not understand the proof: why does the first equality hold? Moreover, g(x) does not depend on k which is strange. Line 129: should be \eta_k = \eta = … and \tau_k = \tau = … line 132: “the left hand side of the rate” – sounds strange. Consider replacement of the word “rate” by inequality. Line 364: it is not a strong convexity. Moreover, it should be clarified explicitly in the main text that strongly convex case analyzed under much stronger assumption than in non-convex case. I understand that authors did this due to the space limitations, but this is a very important aspect that should be mentioned in the main text as well (at least at the high-level). Also, some clarification is needed how to handle the case that the gradient norm is unbounded on the entire space for strongly convex problems. The remark given in lines 370-372 do not address this issue in a proper way (see the comment below). Line 139: the definition of \bar{x}_k does not correspond to the definition used (implicitly) in the proof. Line 142: one should add a remark there that the results for \alpha = 2 can be significantly improved for SGD: instead of the second moment bound one can prove the result with the bound for the variance of the stochastic gradient. Moreover, it seems that one can get it for GClip as well taking into account that very similar result (but in terms of high-probability convergence) was obtained in [11] for restarted GClip and its accelerated variant. Line 170: you have not assumed this for non-convex case Line 173: “coordinate-wise gradients” – should be “coordinate-wise stochastic gradients” instead Line 177-178: “under the Assumption of 2” – sounds unusual Line 182: the definition of Cclip is inaccurate: m_k is undefined, it is not mentioned how to choose the next iterate. I guess, it should be like in GClip but with different coordinate-wise clipping operator. Section 4.1: authors should write explicitly that for these results they do not use smoothness of the function. Also, some clarification is needed how to handle the case that the gradient norm is unbounded on the entire space for strongly convex problems. Line 367: I agree, but it should be explicitly stated that you consider smooth problems. Line 370: “he projected…” → The projected lines 370-372: It is not straightforward: in your proof (the last inequality on the page 13) you assume that the gradient at the solution equals zero which does not hold for any constrained problem. It holds only when the solution lies in the interior of the set where the optimization is considered. These lines should be modified and clarified. Line 375: assumption 1 → Assumption 1. Line 376 (formula below): What is G? Is it G from Assumption 4? It should be clarified. Lines 377-378: the punctuation is incorrect. Lines 379-380: I do not understand the strange symbol appearing in the second inequality in the second row. Is it an indicator function? Moreover, it should be the norm, not the component-wise absolute values of g(x) in the condition defining this indicator. Line 380: I believe, it should be explained more. It does not follow from Markov inequality explicitly. I managed to get this after applying Holder inequality and then Markov inequality. Lines 393-394: the first, the second and the third inequalities should be explained in the text. They are correct, but still sufficient explanation is needed. Lines 405-406: there are too many inaccuracies and missing explanation in this sequence of inequalities. As I mentioned before it is not clear why the first inequality holds and how to fix this to get the correct one. Moreover, I believe it should be -\frac{8\sigma}{3} instead of \frac{8\sigma}{3}.


Review 2

Summary and Contributions: The paper proposes a new clipping based adaptive gradient method. They motivate this clipping based on the observation of heavy-tailed noise of the stochastic gradients. The paper also presents convergence results for the new clipped method under a heavy tail noise type assumption and provide new lower bounds for stochastic gradient algorithm under these assumptions showing that their algorithm is optimal.

Strengths: > The novel lower bounds in combination with the heavy tail noise assumptions. > Clipping algorithm that matches these lower bounds > Interesting storyline around the empirical observation of heavy tail gradient noise

Weaknesses: >The paper has many issues in clarity, making it hard to discern exactly what is being displayed in the histograms of noise, missing assumptions and definitions in the proofs. Below I give details > The idea behind using clipping is not new (AMSgrad and several follow up papers). > The Assumptions for Theorem 4 are very restrictive (no examples are known/given) > It is not clear exactly what settings were used in the experiments.

Correctness: I am not sure, the theoretical statements are not sufficiently clear and nor are the experimental details sufficiently clear. No code was provided to fill in these gaps.

Clarity: the paper is not clear and contains several small mistakes throughout.

Relation to Prior Work: The authors recall much of the related work and contrast their results with others. The additional references are: Adaptive Methods for Nonconvex Optimization, Zaheer et al, Neurips, 2018 RAdam: Liyuan Liu, Haoming Jiang, Pengcheng He, Weizhu Chen, Xiaodong Liu, Jianfeng Gao, and Jiawei Han, “On the variance of the adaptive learning rate and beyond,” AdamW : Ilya Loshchilov and Frank Hutter, “Decoupled weight decay regularization

Reproducibility: No

Additional Feedback: ******** AFTER RESPONSE ******* I appreciate the authors responses. One of my questions that still remains is around the parameter tunings, which I repeat here for convenience. Also, in light of the many technical issues pointed out by Reviewer1, and also because of the many small issues and missing definition I pointed out, I feel the paper needs a major revision, and because of this I have decided to keep my previous score. > Line 229: "we set lr = 1e-4, β1 = 0.9, β2 = 0.99, ε = 1e-5 and wd = 1e-5." I suppose that lr is the learning rate. What is wd? This parameter was not defined for ACClip. Also, in contradiction to this, on line 234 it is stated: > Line 234: "The learning rates and hyperparameters for each method have been extensively tuned to provide best performance on validation set." Given my comment about line 229, I'm not sure what, or how many, hyperparameters ACClip has. It appears it has more hyperparameters than Adam. In which case, it would not be fair comparison to Adam if all these parameter were tuned. ******************** I find the overall idea of the paper interesting. The empirical study of the distribution of the noise, and establishing lower bounds and an algorithm to match. But I see too many issues with clarity and mistakes throughout to recommend publishing. Please see my list below. But as such, I encourage the authors to address these issues, re-consider the presentation of their results, and re-submit. > Discussion around noise ||g(x) -\nabla f(x)|| distribution. For plotting the histogram of noise in Figures 1 and Figures 2, it is not clear exactly how you are sampling the gradients. For Figure 1, I supposed these histograms referred to the distribution of || g(x_k) -\nabla f(x_k)|| for k=1,..., K. But this is not clear. It could be that you are generating this histogram on a fixed x_0 and simply re-sampling the stochastic gradient g(x_0). This should be clarified since it is central to the papers storyline. For Figure 2, the different plots are captioned by the iteration count (0k, 4k, 12k and 36k). Does this mean you fix the x_kth iterate and resample the gradient on this iterate? Or is this the distribution of iterates between the 0k and 4k, 4k and 12k and 12k and 36 k iterate? This distinction is important to understand the meaning of these figures. > Remark 1, Line 116: The proof here has several issues. First, what is x with no subscript? Also ∇f(x_{k}+1) should be ∇f(x_{k+1})? Still the next step doesn't follow. It seems instead you are using x_{k+1} - \eta_k x_k? > Theorem 2. -- Capital K was not defined. I suppose it means you run GClip for a finite horizon of K steps. -- on line 133 you state E∥∇f(x)∥ ≤ O(K^{-(α−1)/(3α−2)} ). Something is missing here. first, it should be the average of the norm gradients or the minimum over the iterates. Second, what is x here? > Theorem 4: -- There is no Assumption 4 in the main text. Same issue in Theorem 5 -- Furthermore, can you provide an example of a smooth, strongly convex functions that also satisfies the bounded noise Assumption 4? I know these assumptions are commonly used, but it is a very restrictive set of assumptions. Note that the difficulty here is that strong convexity implies that the full batch gradient \nabla f(x) is unbounded. > Line 146: By E[|g(x)|^α] ≤ 1 did you mean Assumption 4? Otherwise, why the switch from centered momentum to uncentered momentum? > Line 152: Again G>0 that I suppose should be \sigma>0 > Algorithm 1. Why is \tau_k in ACClip updated in this particular way? This was not clearly motivated in the paper. Also, it appears to be closely related to AMSgrad. Could the authors comment on this? > Figure 3: "Performance of different algorithms for training a toy transformer-XL model described in Section 4." There is no description of this model in Section 4. > Table 2: by "better evaluation loss" do you mean validation loss? If not, what does this mean? Also, the Masked LM accuracy is not defined. > Line 229: "we set lr = 1e-4, β1 = 0.9, β2 = 0.99, ε = 1e-5 and wd = 1e-5." I suppose that lr is the learning rate. What is wd? This parameter was not defined for ACClip. Also, in contradiction to this, on line 234 it is stated: > Line 234: "The learning rates and hyperparameters for each method have been extensively tuned to provide best performance on validation set." Given my comment about line 229, I'm not sure what, or how many, hyperparameters ACClip has. It appears it has more hyperparameters than Adam. In which case, it would not be fair comparison to Adam if all these parameter were tuned. << Minor suggestions>>> > Line 45: we motivated the a novel adaptive-threshold => we motivate the novel adaptive-threshold > Line 102: we assume access to an unbiased stochastic gradient E[g(x)] = ∇f(x,ξ). This should be ∇f(x), no? > Line 105: G>0 is not used. I suppose this is meant to be \sigma >0. This G>0 then makes multiple appearances throughout the paper. > Line 203: A minor note, if alpha =1 gave the best performance for ACClip for all problems, why not simply present ACClip with alpha = 1? > Section 6, in the text you refer to RMSProp, but in the updates it appears to be Adam (since the \epsilon is outside the square root). You even label h_Adam. Supplementary Material: > Assumption 4 is labelled as mu-strong-convexity, but it is a different type of bounded noise assumption. > Line 379, there is a broken symbol in the equations. I think you accidentally used \mathbf{1} which I suppose was meant to be an indicator function.


Review 3

Summary and Contributions: The paper considers heavy-tail noise phenomenon in training deep nets and provides some evidence that clipped SGD and the like are better than vanilla SGD. The topic is important and the results are encouraging.

Strengths: see above

Weaknesses: The empirical analysis in Section 2 is rather weak. It is not substantiated by a careful statistical testing, and hence hard to draw any serious conclusion from a figure. The claim that adaptive methods outperform SGD under heavy-tail noise is not well supported. What shown in Figures 1(a)+(e) are the performance for a single run and for a single parameter selection (not mentioned in the paper). What were the mini-batch sizes used? I checked the supp material but could not find them either. It is not clear what is the sample size in the figure? The noise norm ||g(x) - grad(f(x)|| plot is for a single point $x$ or what? Please give a detailed description of the experiment. The claim that clipped SGD is theoretically better than SGD is not clear either. The paper shows that when all the unknown parameters are given to set the clipping threshold, the clipped SGD converges at a certain rate. But it does not tell us what if SGD is also given such information. The paper also gives an example in Remark 1 claiming that SGD is “non-convergent”, but it is likely that this is false (see below). The experiments are rather orthogonal to the theoretical development in the paper. It would be better if the author(s) started with some problems that can justify the used assumptions. It is okay to run on large deep models, but since those assumptions are not likely to hold, and all considered methods rely heavily on parameter tuning, the performance gain of a single run (e.g., in Figure 3) may have nothing to do with the theory developed in the paper.

Correctness: It seems that Remark 1 is wrong. The function x^2 is deterministic, so I do not understand what is the noise here and how is it possible that the variance of the noise in this case becomes infinity? The claim that || \grad(f(x))||=\infty is wrong too, since \grad(f(x))=x for any x in R. ============================= After response: Thank you for the response! Since my concern on Remark 1 has been clarified, I have updated my rating. Overall, I found the paper interesting both empirically and theoretically. However, I also share similar concerns with R1+R2 that the current version needs to be significantly revised before acceptance.

Clarity: Yes.

Relation to Prior Work: Yes.

Reproducibility: No

Additional Feedback: 1. Please give a more careful description of the problem formulation in Page 3, lines 100-105. What is a “differentiable stochastic function”? Why \xi represents the mini-batches? It should be a RV so that the expectation operator is meaningful, a mini-batch is just a realization of \xi? E[g(x)] should equal \grad{f(x)} and not \grad{f(x, \xi)}. 2. Please replace the L-smoothness assumption in Theorem 4 by a strong convexity one. 3. The complexity given in Remark 3 is misleading. It should be stated in terms of min(E[||grad{f(x_k)}||]) over k \in {1, …, K}; the one in the paper ignores the min. Minor comments: 1. Table 1: (f(x_k)-f(x*) => f(x_k)-f(x*) 2. Footnote 1: text missing 3. Assumption 4 is not about \mu-strong convexity.


Review 4

Summary and Contributions: This paper studies studies the heaviness of stochastic gradients in practical machine learning problems, and how gradient clipping can be used in such situations to give both theoretical and empirical optimal rates of convergence. An adaptive coordinate-wise gradient clipping method is proposed which has very impressive empirical performance.

Strengths: Significance - The analysis in Section 3 is of high significance, as it clearly demonstrates the benefit of clipping in dealing with heavy tailed noise. - The empirical results on BERT training are very strong, and may well influence how these models are trained in the future Novelty - The main novelty comes through the analysis in section 3. This is important work, as it shows that clipping achieves the best possible rate of convergence (up to a constant). - The adaptive clipping technique is novel, but similar to existing methods. The discussion in section 6 neatly ties ACClip to existing methods. Relevance - This work is of high relevance for both theoretical (tight bounds) and empirical (strong performance on BERT training) reasons.

Weaknesses: - A limitation of the paper is that the potential multiplicative nature of the noise is not discussed (i.e. g_k = a \nabla f + b where a and b are random variables). For example, I'd like to see the plots in Figure 1 for ||g-\nabla f|| and ||g-\nabla f|| / ||\nabla f||. That said, it is hard for an 8-page paper to deal with everything, so I think it is fine that this was left out.

Correctness: I did not check carefully, but they seemed correct. One minor point: - Figure 2(b) Iteration 60k. It looks like the Noise norms are double the previous plot. This looks weird - there might be an error here.

Clarity: In general the paper is well written. Below are a few minor changes: - Figure 1 caption: "nosie" -> "noise" - Line 117: Add another step or two to help the reader (its not hard, but you have space in the line, so why not?), e.g. ... \geq \eta_k^2 E[||g(x)||^2] \geq \eta_k^2 E[||g(x) - \nalba f(x)||^2] + ... = \infty - Line 182: Unless I missed something, m_k was not defined earlier. I assume m_k should've been g_k. Also it is not explained how CClip is used in SGD. Again, this is not hard to work out, but writing the update like equation (GClip) would make it more explicit and easier for the reader. - Algorithm 1 ACClip, Line 7: Put "return..." on its own line.

Relation to Prior Work: - The related work section could be expanded to add more work on adaptive clipping, e.g. Pichapati, Venkatadheeraj, et al. "AdaCliP: Adaptive clipping for private SGD." arXiv preprint arXiv:1908.07643 (2019).

Reproducibility: Yes

Additional Feedback:

[Author Response · NeurIPS 2020]

**Reviewer 1:**  We very much thank the reviewer for the excellent and thorough feedback!

**1. "Typos and inaccuracies"**. Minor edits such as awkward wording and incorrect subscripts (e.g. $\eta_k = \eta = ...$ will

be corrected following the reviewer's suggestion. We next address the major technical concerns raised.

**Detailed explanation of remark 1**: Though the function we are optimizing $f(x) = x^2/2$ is deterministic, we

use a **stochastic gradient oracle** $g_k = g(x_k) = \nabla f(x) + \xi = x + \xi$, where $\xi \in \mathbb{R}^d$ is a random variable with

$\mathbb{E}\|\xi\|^2 = \infty, \mathbb{E}\|\xi\|^\alpha = \sigma^\alpha, \mathbb{E}[\xi] = \vec{0}$. It is the noise in this gradient oracle which causes divergence. Specifically,

$\mathbb{E}[\|\nabla f(x_{k+1})\|^2] = \mathbb{E}[\|x_{k+1}\|^2] = \mathbb{E}\|x_k - \eta_k g_k\|^2 = \mathbb{E}\|x_k - \eta_k(x_k + \xi)\|^2 = \mathbb{E}\|(1 - \eta_k)x_k - \eta_k\xi\|^2 = \mathbb{E}\|(1 -$

$\eta_k)x_k\|^2 - 2(1 - \eta_k)\eta_k x_k^\top \mathbb{E}[\xi] + \eta_k^2 \mathbb{E}\|\xi\|^2 \geq \eta_k^2 \mathbb{E}\|\xi\|^2 = \infty$. Note that this holds for **any** fixed $\eta_k > 0$ even if allowed

to depend on the statistics of the noise distribution (such as $\sigma$ or $\alpha$).

**Detailed explanation of line 405:** We agree with the reviewer that more details and minor corrections are needed

in the appendix, but our claims remain unaffected. We hope the reviewer, if convinced by the derivation below,

could confirm the correctness of our proof in the discussion. The intuition for case 2 is that when gradient is very

large($\tau = \mathcal{O}(\eta^{-1/\alpha}) = \mathcal{O}(K^{1/(3\alpha-2)})$), noise gets dominated by the gradient. Recall that $\|\nabla f(x_k)\| > \tau/2$,

$\tau = \sigma(\eta L)^{-1/\alpha}$, $p = \mathbb{P}\{\|g_k\| \leq \tau\}$, and that expectation is taken wrt $g_k$. We use $\nabla f$ as a shorthand for $\nabla f(x_k)$:

$$\mathbb{E}[\langle \nabla f, g_k\rangle \mathbb{1}_{\{\|g_k\|\leq\tau\}}] \geq \mathbb{E}[(\|\nabla f\|^2 - \|\nabla f\|\|g_k - \nabla f\|)\mathbb{1}_{\{\|g_k\|\leq\tau\}}]$$

$$\geq \mathbb{E}[\|\nabla f\|^2 \mathbb{1}_{\{\|g_k\|\leq\tau\}} - \tfrac{1}{2}\|\nabla f\|^2 \mathbb{1}_{\{\|g_k\|\leq\tau, \|g_k-\nabla f\|\leq\tau/4\}} - \|\nabla f\|\|g_k - \nabla f\|\mathbb{1}_{\{\|g_k\|\leq\tau, \|g_k-\nabla f\|\geq\tau/4\}}]$$

$$\geq \tfrac{p}{2}\|\nabla f\|^2 - \|\nabla f\|\mathbb{E}[\|g_k - \nabla f\|\mathbb{1}_{\{\|g_k-\nabla f\|\geq\tau/4\}}] \geq \tfrac{p}{2}\|\nabla f\|^2 - \|\nabla f\|\tfrac{\sigma^\alpha}{(\tau/4)^{\alpha-1}}$$

The first inequality uses $\langle \nabla f, g_k\rangle = \|\nabla f\|^2 + \langle \nabla f, g_k - \nabla f\rangle$. The second line follows by $\|\nabla f\| > \tau/2$ and $\|g_k -$

$\nabla f\| < \tau/4 \implies -\|\nabla f\|\|g_k - \nabla f\| \geq -\|\nabla f\|^2/2$. Then, we use the fact that $\mathbb{P}(A \cap B) \leq \mathbb{P}(A)$ for any random

events $A$ and $B$. The last inequality follows by $\sigma^\alpha \geq \mathbb{E}[\|g_k - \nabla f\|^\alpha] \geq \mathbb{E}[\|g_k - \nabla f\|(\tfrac{\tau}{4})^{\alpha-1}\mathbb{1}_{\|g_k-\nabla f\|\geq\tau/4}]$. With

the above, we go back to line 405,

$$\mathbb{E}[\langle \nabla f, \hat{g}_k\rangle] = \mathbb{E}[\langle \nabla f, g_k\rangle \mathbb{1}\{\|g_k\| \leq \tau\}] + \mathbb{E}[\langle \nabla f, g_k/\|g_k\|\rangle \mathbb{1}\{\|g_k\| \geq \tau\}]$$

$$\geq \tfrac{p}{2}\|\nabla f\|^2 - \|\nabla f\|\tfrac{\sigma^\alpha}{(\tau/4)^{\alpha-1}} + (1 - p)\|\nabla f\|/3 - \tfrac{8}{3}\mathbb{E}[\|\nabla f - g_k\|]$$

$$\geq \|\nabla f\|/3 - \|\nabla f\|\tfrac{\sigma^\alpha}{(\tau/4)^{\alpha-1}} - \tfrac{8\sigma}{3} \geq \|\nabla f\|/3 - \|\nabla f\|/12 - \|\nabla f\|/6$$

The second line follows by Lemma 11. Since $\tau \geq 2$ (informally, recall $\tau = \mathcal{O}(\eta^{-1/\alpha}) = \mathcal{O}(K^{1/(3\alpha-2)})$ increases

with total steps $K$), and $\|\nabla f\| \geq \tau/2$, we have $\tfrac{p}{2}\|\nabla f\|^2 \geq p\|\nabla f\|/3$. Then, by imposing $\eta \leq \frac{1}{L(48\sigma)^{\alpha/(\alpha-1)}}$ and since

$\tau = \sigma(\eta L)^{-1/\alpha}$ by defn., we have $\frac{\sigma^\alpha}{(\tau/4)^{\alpha-1}} \leq \frac{1}{12}$. Finally, combining with line 407 we gethe desired result in the last

inequality above. Note that the proof above corrects some mistakes and hence differs slightly from the submission.

**2. "Strongly convex and bounded second moment implies bounded domain".** This is true and is the classical

setting for all stochastic subgradient methods under strongly convex assumption (e.g. Chapter 6.1 Bubeck, Sébastien.

"Convex optimization: Algorithms and complexity. "). Since this is a standard setting, we briefly discussed the bounded

domain in line 369. We will move these comments to Thm 4 as the reviewer suggested.

**Reviewer 2:**  We thank the reviewer for detailed comments. We address the reviewer's question as follows and

will edit the paper accordingly. **0. "Novelty of Clipping"** As reviewer said, the idea of clipping is not new but

the understanding has been heuristic (to avoid gradient explosion). We provide solid theoretical justification for its

advantage by formalizing clipping's faster convergence under the experiment motivated by heavy-tailed condition. This

help explain the mismatch between theory (SGD is optimal; adaptive methods give worse convergence bound) and

practice (adaptive methods converge much faster in many settings). **1. "Experimental setting of Figure 1":** We fix

the model parameters and keep sampling minibatches without updating the model (see line 80) to plot Figure 1(b)(f).

**2. "Remark 1":** Please see point 1 in response to Reviewer 1. **3. "Thm 2":** $K$ is the number of iterations, and $x_k$ is

the variable at iteration $k$. Please take a look at line 130. The convergence is in the average sense as we are summing

up all the iterate and then divide by $K$. **4. "Thm 4":** Assump. 4 assumes strong convexity and is in the appendix

due to limited space. We will remove some experiments from the main part and move the optimization setting (hence

clarifying Figure 1) to the main text. See the "explanation of line 405" in response to reviewer 1 for more proof details.

**5. "Line 146":** $\mathbb{E}[|g(x)|^\alpha] \leq 1$ refers to the function that we used to prove the lower bound. In other words, the

counter-example presented in appendix F satisfies $\mathbb{E}[|g(x)|^\alpha] \leq 1$. Note that this is a stricter condition than that used

by our upper-bound, making our lower bound even stronger. **6. "Update of the threshold":** We use the exponential

moving average estimator motivated by ADAM, which also uses it to estimate moments. **7. "Other":** Evaluation loss =

validation set, and wd = weight decay. Hence, equal number of parameters are tuned for ADAM and ACClip.

**Reviewer 3:**  1. We used the statistical estimator studied in [Simsekli, et al ICML 2018]. Results are in Figure 8. 2.

Please refer to Reviewer 2 point 1 for experiment setup. 3. Please refer to Reviewer 1 point 1 for proof of remark 1.

**Reviewer 4:**  We appreciate the reviewer's comment. We will address the typos and further improve readability by

incorporating all reviewer's comments.

[Meta-Review · NeurIPS 2020]

There was a reasonable amount of discussion about this paper. The author feedback clarified a variety of issues which caused some reviewers to increase their scores, while some of the discussion caused other reviewers to decrease their scores. Although there was one holdout, the majority of the reviewers leaned towards rejection of the paper. However, I believe this is one of the rare cases where the AC should recommend for the PC to accept the paper against the recommendations of the reviewers. (And to be clear, I'm not doing this lightly: among the reviewers recommending rejection are optimization experts/stars who I greatly respect.) The main reason I'm recommending acceptance is due to the broader context and potential impact of the paper. The paper is giving new insights into one of the most important/mis-understood issues in the *practice* of optimization for machine learning. It is also providing theory under non-standard assumptions to explain empirical observations that were not explained by previous theory, and in my opinion gives a creative solution to the observed issues (even the new algorithm isn't dramatically different than existing methods). The number of people that could be helped by a better understanding of why Adam-related and clipping-related methods can/should be use to train language models is simply *much* bigger than the optimization community (more classes around the world teach LSTMs than teach strong-convexity). I do not think that the paper should be rejected because of minor issues related to things we view as important in the optimization community, given the large potential impact of the work. That being said, if the PCs agree with me then there are two things I want to see from the authors in preparation of the camera-ready version (in addition to addressing the other review comments): 1. Fix the issues associated with making the "strong-convexity and gradient bounded assumption", which the reviewers correctly point out cannot be true for an entire real space. The standard/preferred correction to this issue is to add projection step onto a compact/convex set to the algorithm. Another fix is to assume or prove that the iterates all stay inside a set. Proving this would be strongly preferred since I'm not sure we should make this assumption under the heavy-tailed noise assumption. (The reasons I don't view this as a crucial issue: this assumption combination is unfortunately pretty standard, and I think the paper would have been reviewed more favourably if the results relying on this combination simply hadn't been included.) 2. Two of the reviewers pointed out potential mistakes in the analysis, which were addressed in the author response. However, the authors should be careful to re-write the analysis as clearly as possible. Further, I think the authors should share their work with several optimization-expert colleagues and get them to check the analysis carefully. If critical issues are found, I hope that the authors do the right thing and withdraw the submission.